# Sampling motion trajectories during hippocampal theta sequences

**Balazs B Ujfalussy[1,2]\*[†], Gergő Orbán[3†]**

[1]Laboratory of Biological Computation, Institute of Experimental Medicine, Budapest, Hungary; [2]Laboratory of Neuronal Signalling, Institute of Experimental Medicine, Budapest, Budapest, Hungary; [3]Computational Systems Neuroscience Lab, Wigner Research Center for Physics, Budapest, Budapest, Hungary

**Abstract** Efficient planning in complex environments requires that uncertainty associated with current inferences and possible consequences of forthcoming actions is represented. Representation of uncertainty has been established in sensory systems during simple perceptual decision making tasks but it remains unclear if complex cognitive computations such as planning and navigation are also supported by probabilistic neural representations. Here, we capitalized on gradually changing uncertainty along planned motion trajectories during hippocampal theta sequences to capture signatures of uncertainty representation in population responses. In contrast with prominent theories, we found no evidence of encoding parameters of probability distributions in the momentary population activity recorded in an open-field navigation task in rats. Instead, uncertainty was encoded sequentially by sampling motion trajectories randomly and efficiently in subsequent theta cycles from the distribution of potential trajectories. Our analysis is the first to demonstrate that the hippocampus is well equipped to contribute to optimal planning by representing uncertainty.

## Editor's evaluation

This paper will be of interest to neuroscientists interested in predictive coding and planning. It presents a novel analysis of hippocampal place cells during exploration of an open arena. It performs a comprehensive comparison of real and synthetic data to determine which encoding model best explains population activity in the hippocampus.

**\*For correspondence:**
ujfalussy.balazs@koki.hu

[†]Co-senior author

**Competing interest:** The authors declare that no competing interests exist.

## Introduction

Model-based planning and predictions are necessary for flexible behavior in a range of cognitive tasks. In particular, navigation is a domain that is ecologically highly relevant not only for humans but for rodents as well, which established a field for parallel investigation of the theory of planning, the underlying cognitive computations, and their neural underpinnings (*Hunt et al., 2021*; *Mattar and Lengyel, 2022*). Importantly, predictions extending into the future have to cope with uncertainty coming from multiple sources: uncertainty in the current state of the environment (our current location relative to a dangerous spot, the satiety of a predator or the actual geometry of the environment) and the availability of multiple future options when evaluating upcoming choices (*Glimcher, 2003*; *Redish, 2016*). Whether and how this planning-related uncertainty is represented in the brain is not known.

The hippocampus has been established as one of the brain areas critically involved in both spatial navigation and more abstract planning (*O'Keefe and Nadel, 1978*; *Miller et al., 2017*). Recent progress in recording techniques and analysis methods largely contributed to understanding of the neuronal mechanisms underlying such computations (*Pfeiffer and Foster, 2013*; *Kay et al., 2020*). A crucial insight gained about the neural code underlying navigation is that neuron populations in

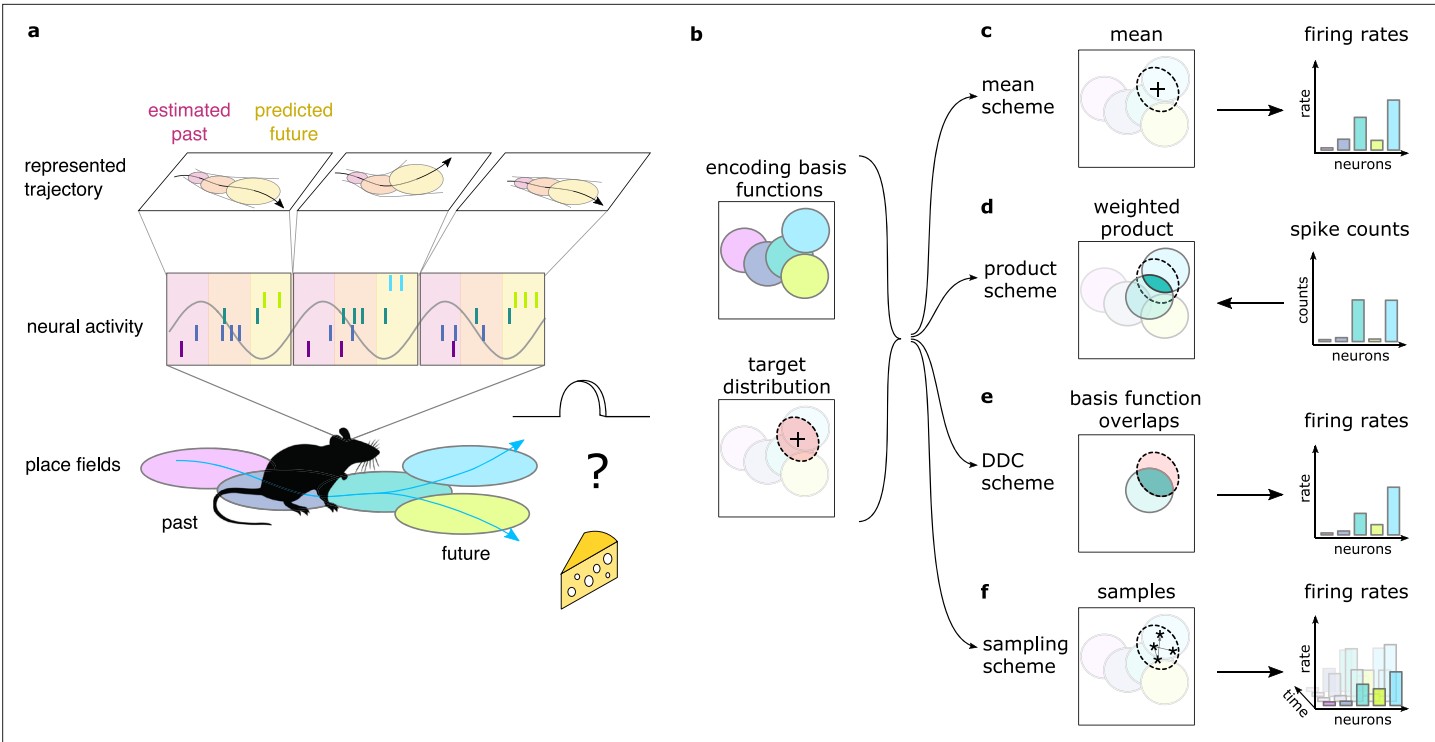

**Figure 1.** Theta sequences, uncertainty and variability. (**a**) Schematic showing the way hippocampal place cell activity represents possible trajectories in subsequent theta cycles during navigation. (**b-f**) Schemes for representing a target probability distribution (b, bottom) by the activity of a population of neurons each associated with an encoding basis function (related to their place field, Methods; b, top). (**c**) In the mean scheme the firing rates of the neurons (right, colored bars) are defined as the value of their basis functions at the mean of the target distribution (left, cross). (**d**) Similar to other schemes, the product scheme also defines a mapping between the probability distribution and the activity of the neuron population but it is easier to understand this mapping in the reverse direction (arrow): from the spike counts (right) to the represented distribution (left). The spike count of each neuron (right) can be considered as its vote for the contribution of their tuning curves to the represented distribution and ultimately the target distribution is approximated as the weighted product of these tuning curves (left). (**e**) In the DDC scheme the firing rate of each neuron (right) is defined by the overlap between the target distribution and the basis function (left). (**f**) In the sampling scheme the firing rate of the neurons at each point in time (right) equals the value of their basis functions at the location sampled from the target distribution (left, asterisks).

the hippocampus represent the trajectory of the animal on multiple time scales: Not only the current position of the animal can be read out at the behavioral time scale (*O'Keefe and Nadel, 1978*; *Wilson and McNaughton, 1993*), but also trajectories starting in the past and ending in the near future are repeatedly expressed on a shorter time scale at accelerated speed during individual cycles of the 6–10 Hz theta oscillation (theta sequences, *Foster and Wilson, 2007*, *Figure 1a*). Moreover, features characteristic of planning can be identified in the properties of encoded trajectories, such as their dependence on the immediate context the animal is in, and on the span of the current run, future choices and rewards (*Johnson and Redish, 2007*; *Gupta et al., 2012*; *Wikenheiser and Redish, 2015*; *Tang et al., 2021*; *Zheng et al., 2020*). These data provide strong support for a computational framework where planning relies on sequential activity patterns in the hippocampus delineating future locations based on the current beliefs of the animal (*Stachenfeld et al., 2017*; *Miller et al., 2017*). Whether hippocampal computations also take into account the uncertainty associated with planning and thus the population activity represents the uncertainty of the encoded trajectories has not been studied yet.

Neuronal representations of uncertainty have been extensively studied in sensory systems (*Ma et al., 2006*; *Orbán et al., 2016*; *Vértes and Sahani, 2018*; *Walker et al., 2020*). Schemes for representing uncertainty fall into three broad categories (*Figure 1d–f*). In the first two categories (*product* and Distributed Distributional Code, *DDC*, schemes), the firing rate of a population encodes a complete probability distribution over spatial locations *instantaneously* at any given time by representing the parameters of the distribution (*Wainwright and Jordan, 2007*, Methods). In the *product scheme*, the firing rate of neurons encode a probability distribution through taking the product of

the tuning functions (basis functions, Methods) of the coactive neurons (*Ma et al., 2006*; *Pouget et al., 2013*, *Figure 1d*). In contrast, in the *DDC scheme* a population of neurons represents a probability distribution by signalling the overlap between the distribution and the basis function of individual neurons (*Zemel et al., 1998*; *Vértes and Sahani, 2018*, *Figure 1e*). In the third category, the *sampling scheme*, the population activity represents a single value sampled stochastically from the target distribution. In this case uncertainty is represented *sequentially* by the across-time variability of the neuronal activity (*Fiser et al., 2010*, *Figure 1f*). These coding schemes provide a firm theoretical background to investigate the representation of uncertainty in hippocampus. Importantly, all these schemes have been developed for static features, where the represented features do not change in time (*Ma et al., 2006*; *Orbán et al., 2016*; but see *Kutschireiter et al., 2017*). In contrast, trajectories represented in the hippocampus encode the temporally changing position of the animal. Here, we extend the coding schemes to be able to accommodate the encoding of uncertainty associated with dynamic motion trajectories and investigate their neuronal signatures in rats while navigating an open environment.

Probabilistic planning in an open field entails the representation of subjective beliefs about the past, present, and future states, which requires that a continuous probability distribution over possible locations is represented. Previous studies investigated prospective codes during theta sequences in a constrained setting, in which binary decisions were required in a spatial navigation task (*Johnson and Redish, 2007*; *Kay et al., 2020*; *Tang et al., 2021*). These studies found that alternative choices were encoded sequentially in distinct theta cycles suggesting a sampling based representation. However, the dominant source of future uncertainty in these tasks is directly associated with the binary choice of the animal (left or right) and it remains unclear whether this generalizes to other sources of uncertainty relevant in open field navigation. In particular, it has been widely reported that the hippocampal spatial code has different properties in linear tracks, where the physical movement of the animal is constrained by the environment, than during open-field navigation (*Buzsáki, 2005*). Moreover, these previous studies did not attempt to test the consistency of the hippocampal code with alternative schemes for representing uncertainty. Thus, the way hippocampal populations contribute to probabilistic planning during general open-field navigation remains an open question.

In the present paper, we propose that the hippocampus is performing probabilistic inference in a model that represents the temporal dependencies between spatial locations. Using a computational model, we demonstrate that key features of the hippocampal single neuron and population activity are compatible with representing uncertainty of motion trajectories in the population activity during theta sequences. Further, by developing novel computational measures, we pitch three alternative schemes of uncertainty representation and a scheme that lacks the capacity to represent uncertainty, the *mean* scheme (*Figure 1c*), against each other and demonstrate that hippocampal activity does not show the hallmarks of schemes encoding probability distributions instantaneously. Instead, we demonstrate that the large and structured trial to trial variability between subsequent theta cycles is consistent with stochastic sampling from potential future trajectories but not with a scheme ignoring the uncertainty by representing only the most likely trajectory. Finally we confirm previous results in simpler mazes by showing that the trajectories sampled in subsequent theta cycles tend to be anti-correlated, a signature of efficient sampling algorithms. These results demonstrate that the brain employs probabilistic computations not only in sensory areas during perceptual decision making but also in associative cortices during naturalistic, high-level cognitive processes.

## Results
### Neural variability increases within theta cycle
A key insight of probabilistic computations is that during planning uncertainty increases as trajectories proceed into more distant future (*Murphy, 2012*; *Sezener et al., 2019*). As a consequence, if planned trajectories are encoded in individual theta sequences, the uncertainty associated with the represented positions increases within a theta cycle (*Figure 1a*). This systematic change in the uncertainty of the represented position during theta cycles is a crucial observation that enabled us to investigate the neuronal signatures of uncertainty during hippocampal theta sequences. For this, we analyzed a previously published dataset (*Pfeiffer and Foster, 2013*). Briefly, rats were trained to collect food reward in a 2×2 m large open arena from one of the 36 uniformly distributed food wells alternating

between random foraging and spatial memory task (*Pfeiffer and Foster, 2013*). Position of the animal was recorded via a pair of distinctly coloured head-mounted LED light. Goal directed navigation in an open arena requires continuous monitoring and online correction of the deviations between the intended and actual motion trajectories. While sequences during both sharp waves and theta oscillations have been implicated in planning, here we focused on theta sequences as they are more strongly tied to the current position and thus averaging over many thousands of theta cycles can provide the necessary statistical power to identify the neuronal correlate of uncertainty representation.

Activity of hippocampal neurons was recorded by 20 tetrodes targeted to dorsal hippocampal area CA1 (*Pfeiffer and Foster, 2013*). Individual neurons typically had location-related activity (*Figure 2b*, see also *Figure 4—figure supplement 1a*), but their spike trains were highly variable (*Skaggs et al., 1996*; *Fenton and Muller, 1998*, *Figure 2a*). We used the empirical tuning curves (i.e., place fields) and a Bayesian decoder (*Zhang et al., 1998*) to estimate the represented spatial position from the spike trains of the recorded population in overlapping 20ms time bins. Theta oscillations partition time into discrete segments and analysis was performed in these cycles separately (*Figure 2c*). Despite the large number of recorded neurons (68–242 putative excitatory cells in 8 sessions from 4 rats), position decoding had a limited accuracy (Fisher lower bound on the decoding error in 20ms bins: 16–30 cm vs. typical trajectory length ~20 cm). Yet, in high spike count cycles we could approximately reconstruct the trajectories encoded in individual theta cycles (*Figure 2c*). We then compared the reconstructed trajectories to the actual trajectory of the animal. We observed substantial deviation between the decoded trajectories and the motion trajectory of the animal: decoded trajectories typically started near the actual location of the animal and then proceeded forward (*Foster and Wilson, 2007*) often departing in both directions from the actual motion trajectory (*Kay et al., 2020*; *Figure 2c*).

To systematically analyse how this deviation depends on the theta phase, we sorted spikes into three classes (early, mid and late). For any given class, we decoded position from the spikes and compared it to the position of the animal shifted forward and backward along the motion trajectory. Time shift dependence of the accuracy of the decoders reveals the most likely portion of the trajectory the given class encodes (*Figure 2d*, see also *Figure 4—figure supplement 2*). For early spikes, the minimum of the average decoding error was shifted backward by ~100ms, while for late spikes +500ms forward shift minimized the decoding error. Note that the position identified by the minima can only establish positions relative to the LED used to record the position of the animal. The relative shifts in the minima of the decoding error across different phases of theta are not affected by the arbitrariness of the positioning of the LED sources. The observed systematic biases indicate that theta sequences tended to start slightly behind the animal and extended into the near future (*Foster and Wilson, 2007*; *Gupta et al., 2012*).

Further, the minima of the decoding errors showed a clear tendency: position decoded from later phases displayed larger average deviation from the motion trajectory of the animal (*Figure 2d*). At the single neuron level, increased deviations at late theta phases resulted in the expansion of place fields (*Skaggs et al., 1996*): Place fields were more compact when estimated from early-phase spikes than those calculated from late-phase activity (*Figure 2e*). At the population level, larger deviation between the motion trajectory and the late phase-decoded future positions can be attributed to the increased variability in encoded possible future locations. Indeed, when we aligned the decoded locations relative to the current location and motion direction of the animal, we observed that the spread of the decoded locations increased in parallel with the forward progress of their mean within theta cycles (*Figure 2f, g*). Taken together, this analysis demonstrated that the variability of the decoded positions is larger at the end of the theta cycle when encoding more uncertain future locations than at the beginning of the cycle when representing the past position.

The observed increase of the variability across theta cycles could be a direct consequence of encoding variable two-dimensional trajectories (*Figure 4—figure supplement 2a, b*) or it may be a signature of the neuronal representation of uncertainty. In the next sections, we set up a synthetic dataset to investigate the theta sequences and their variability in the four alternative coding schemes.

## Synthetic data: testbed for discriminating the encoding schemes

To analyse the distinctive properties of the different coding schemes and to develop specific measures capable of discriminating them, we generated a synthetic dataset in which both the behavioral component and the neural component could be precisely controlled. The behavioral component was

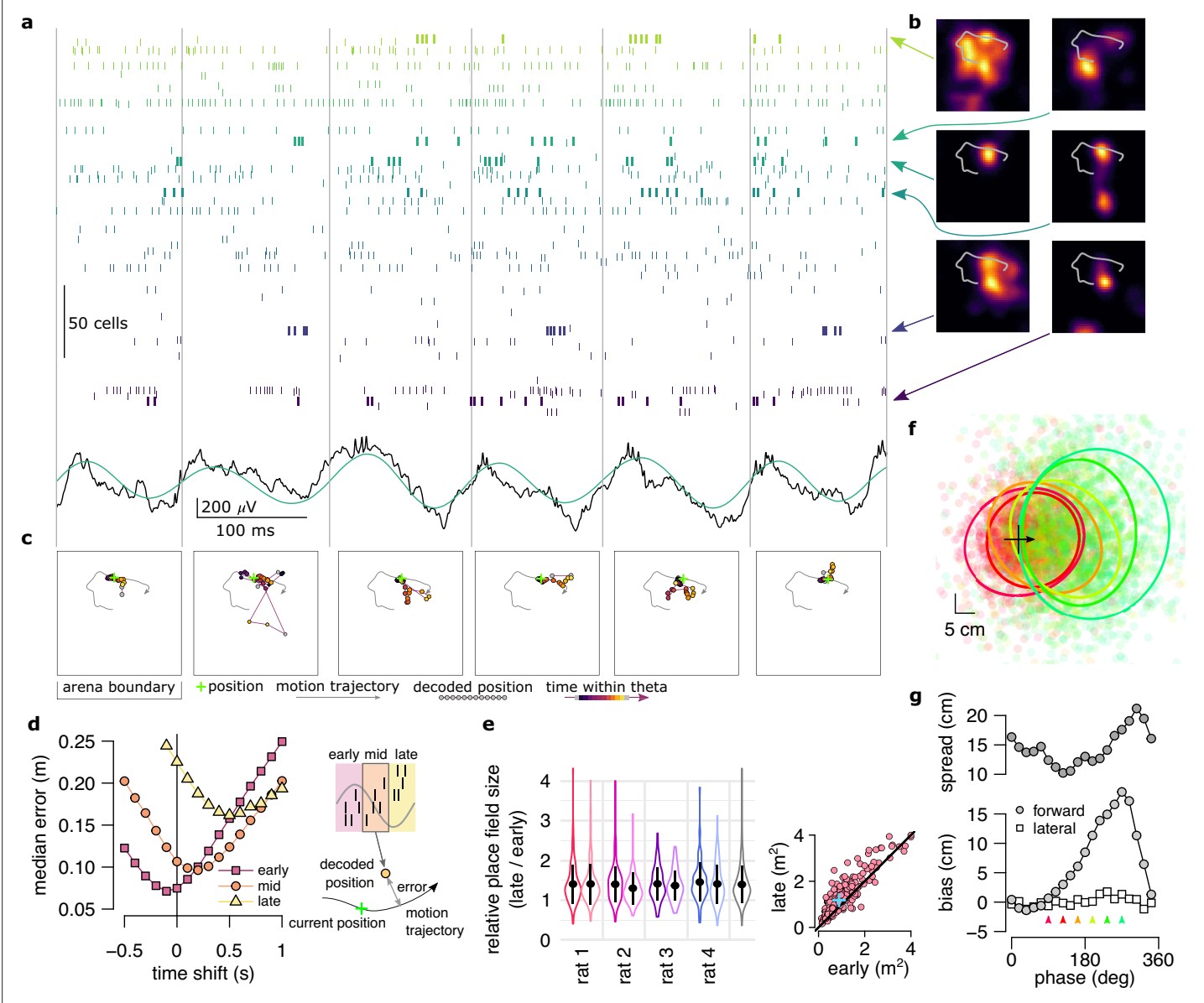

**Figure 2.** Neural variability increases within theta cycle. (**a**) Example spiking activity of 250 cells (top) and raw (black) and theta filtered (green) local field potential (bottom) for 6 consecutive theta cycles (vertical lines). (**b**) Place fields of 6 selected cells in a 2x2 m large open arena. Gray line indicates the motion trajectory during the run episode analysed in a-c. (**c**) Position decoded in overlapping 20ms time bins with 5ms shift (circles) during the 6 theta cycles shown in a. Time within theta cycle is color coded, gray indicates bins on the border of two cycles. Gray line shows the motion trajectory during the run episode, green crosses indicate the location of the animal in each cycle. (**d**) Decoding error for early, mid and late phase spikes (inset, top) calculated as a function of the time shift of the animal's position along its motion trajectory (inset, right). For the analysis in panels d-e each theta cycle was divided into 3 parts with equal spike counts. (**e**) Relative place field size in late versus early theta spikes for the eight sessions (error bars: SD across n=80-263 cells). Gray bar: average and SD across all sessions (n=1264 cells). Inset: Place field size (ratemap area above 10% of the max firing rate) estimated from late vs. early theta spikes in an example session (individual dots correspond to individual place cells, blue cross: median). Only putative excitatory cells are included. To estimate the ratemaps, we shifted the reference positions with $\Delta t$ that minimized decoding error for the given theta phase (see panel d). (**f**) Decoded positions (dots, in 20 ms bins with 5 ms shift) relative to the instantaneous position and motion direction (cross), and 0.5 confidence interval (CI) ellipses for six different theta phases (color, as in panel g). (**g**) Bias (bottom, mean of the decoded positions) and spread (top, see Methods) of decoded positions as a function of theta phase for an example session. Panels d,f,g show data from theta cycles with the highest 10% of spike counts.

matched to the natural movement statistics of rats during navigating a 2D environment. The neural component was constructed such that it could accommodate the alternative encoding schemes for prospective representations during navigation.

Our synthetic dataset had three consecutive levels. First, we simulated a planned trajectory for the rat in a two dimensional open arena by allowing smooth changes in the speed and motion direction (Methods, *Figure 3—figure supplement 1a, b*). Second, similar to the real situation, the simulated animal did not have access to its true position, $x_n$, in theta cycle $n$, but had to infer it from the sensory inputs it had observed in the past, $y_{1:n}$ (Methods). To perform this inference and predict its future position, the animal used a model of its own movements in the environment. In this dynamical generative model, the result of the inference is a posterior distribution over possible trajectories starting $n_p$

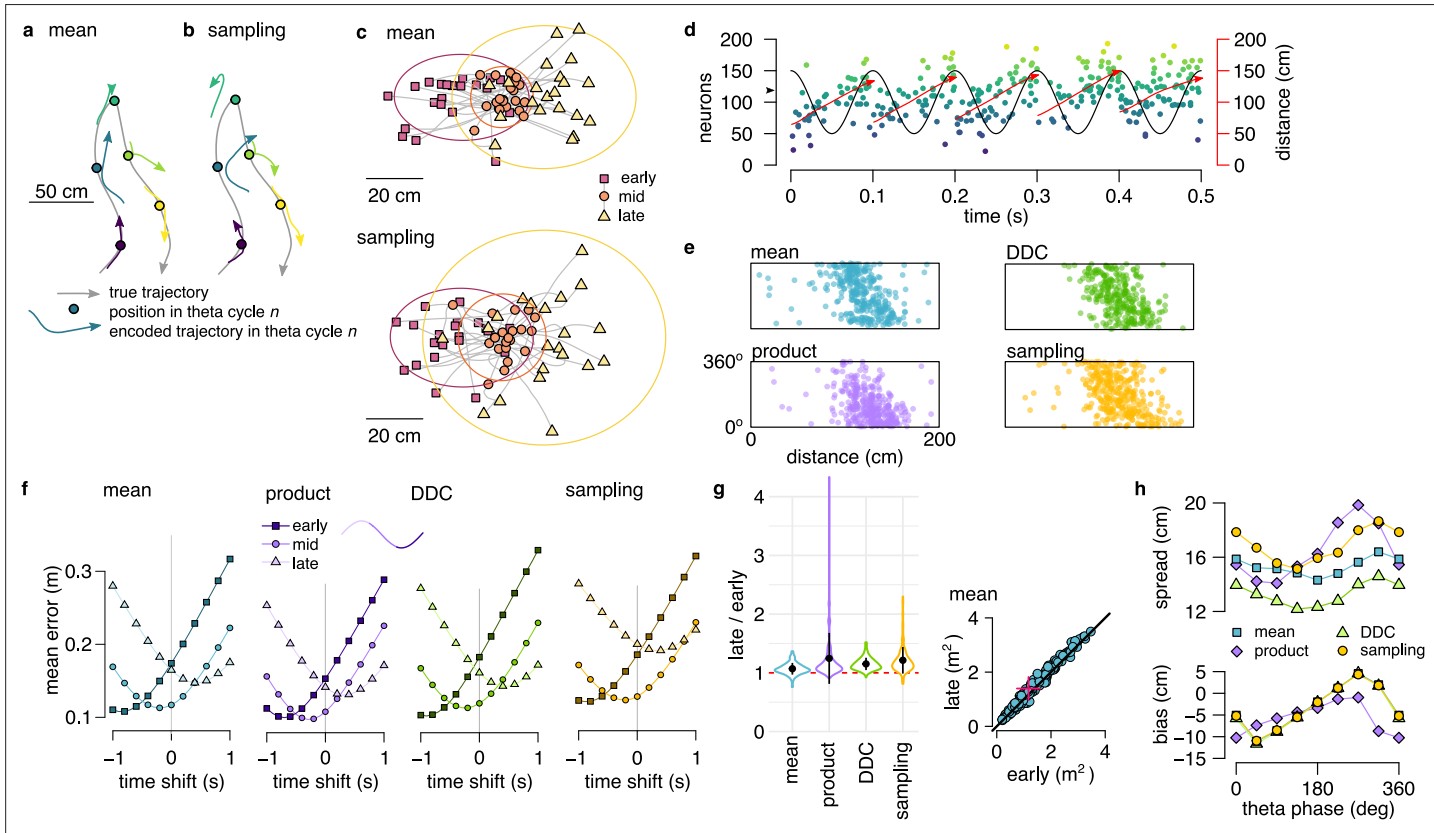

**Figure 3.** Theta sequences in simulated data. (**a**) Motion trajectory of the simulated rat (gray, 10 s) together with its own inferred and predicted most likely (mean) trajectory segments (colored arrows) in five locations separated by 2 s (filled circles). These trajectories were represented in theta cycles using one of the four alternative schemes. (**b**) Same as panel a, but trajectories sampled from the posterior distribution. (**c**) Example represented trajectories aligned to the instantaneous position and direction of the simulated animal in the mean (top) and in the sampling (bottom) scheme. Ellipses indicate 50% CI of all theta cycles. Color code indicates start (red), mid (orange) and end (yellow) of the trajectories. (**d**) Simulated activity of 200 place cells sorted according to the location of their place fields on a linear track (200x10 cm) during an idealized 10 Hz theta oscillation using the mean encoding scheme. Red lines show the 1-dimensional trajectories represented in each theta cycle. Note that the overlap between trajectories is larger here than in panels a-b, because, for clarity, only trajectories at every 20th theta cycle are shown there. (**e**) Theta phase of spikes of an example simulated neuron (arrowhead in panel d) as a function of the animal's position in the four coding schemes. (**f**) Decoding error from early, mid and late phase spikes (highest 5% spike count cycles) as a function of the time shift of the simulated animal's position in a mean, product, DDC and sampling schemes. (**g**) Relative place field size in late versus early theta spikes for the four different encoding schemes (error bars: SD over 200 cells). Inset: place field size estimated from late vs. early theta spikes in the mean scheme. Median is indicated with red cross. (**h**) Decoding bias (bottom) and spread (top) as a function of theta phase for the four different encoding schemes. Decoding was performed in 120° bins with 45° shifts. All theta cycles are included in this analysis, as focusing on the highest spike count cycles highly influences these quantities in the product model.

The online version of this article includes the following figure supplement(s) for figure 3:

**Figure supplement 1.** Inference and movement in the generative model.

**Figure supplement 2.** Comparison of the motion profile of the simulated animal and one of the analysed experimental sessions.

**Figure supplement 3.** Place cell firing in the synthetic and in the experimental data.

steps back in the past and ending $n_f$ steps forward in the future. To generate the motion trajectory of the animal noisy motor commands were calculated from the difference between its planned trajectory and its inferred current position (Methods, *Figure 3—figure supplement 1a, b*). The kinematics was matched between simulation and experimental animals (*Figure 3—figure supplement 2*).

Third, in our simulations the hippocampal population activity encoded an inferred trajectory at an accelerated speed in each theta cycle such that the trajectory started in the past at the beginning of the simulated theta cycle and led to the predicted future states (locations) by the end of the theta cycle (*Figure 3a–c*). To approximately match the size of the synthetic and experimental data, we simulated the activity of 200 hippocampal pyramidal cells (*place cells*). Firing rates of pyramidal cells depended on the encoded spatial location using either of the four different coding schemes (Methods): In the *mean* code, the population encoded the single most likely trajectory without representing uncertainty. In *product* and *DDC* schemes, a snapshot of the population activity at any given theta phase encoded the estimated past or predicted future part of the trajectory in the form of a probability distribution (Methods). Finally, in the sampling scheme, in each theta cycle a single trajectory was sampled stochastically form the distribution of possible trajectories (*Figure 3—figure supplement 1*). Spikes were generated from the firing rates independently across neurons via an inhomogenous Poisson process. The posterior distribution was updated in every ~100ms matching the length of theta cycles. Importantly, all of the four encoding schemes yielded single neuron and population activity dynamics consistent with the known features of hippocampal population activity including spatial tuning, phase precession (*Figure 3d*) and theta sequences (*Figure 3e*, see also *Figure 3—figure supplement 3*).

After generating the synthetic datasets, we investigated how positional information is represented during theta sequences in each of the four alternative coding schemes. We decoded the synthetic population activity in early, mid and late theta phases and compared the estimated position with the actual trajectory of the simulated animal. The deviation between the decoded position and the motion trajectory increased throughout the theta cycle irrespective of the coding scheme (*Figure 3f*) due to the combined effects of the divergence of possible future trajectories and the increased variability of the encoded locations. Moreover, place fields were significantly larger when estimated from late than early-phase spikes in all four encoding schemes (*Figure 3g*, $P < 10^{-16}$ for each scheme). Finally, when aligned to the current position and motion direction of the simulated animal, the spread of the decoded locations increased with the advancement of their mean within theta cycles in all four coding schemes (*Figure 3h*).

Our analyses thus confirmed that the disproportionate increase in the variability of the encoded locations at late theta phase is consistent with encoding trajectories in a dynamical model of the environment irrespective of the representation of the uncertainty. These synthetic datasets provide a functional interpretation of the hippocampal population activity during theta oscillations and offer a unique testbed for contrasting different probabilistic encoding schemes. In the following sections, we will identify hallmarks for each of the encoding schemes of uncertainty and use these hallmarks to discriminate them through the analysis of neural data (*Figure 4a*).

## Testing the product scheme: gain

To discriminate the product scheme from other representations, we capitalize on the specific relationship between response intensity of neurons and uncertainty of the represented variable. In a product representation, a probability distribution over a feature, such as the position, is encoded by the product of the neuronal basis functions (Methods). When the basis functions are localized, as in the case of hippocampal place fields, the width of the encoded distribution tends to decrease with the total number of spikes in the population (*Ma et al., 2006*, *Figure 4b*, Appendix 1, *Figure 4—figure supplement 1*). Therefore, we propose using the systematic decrease of the population firing rate (*gain*) with increasing uncertainty as a hallmark of the product scheme.

We first tested for the specificity of the co-modulation of the population gain with uncertainty to the product scheme: we compared gain modulation in the four different coding schemes in our synthetic dataset. In each of the coding schemes, we identified the theta phase with the maximal uncertainty by the maximal forward bias in encoded positions. For this, we decoded positions from spikes using eight overlapping 120° windows in theta cycles and defined the end of the theta cycles based on the maximum of the average forward bias (*Figure 4C* top). Then we calculated the average number of spikes in a given 120° window as a function of the theta phase. We found that the product scheme

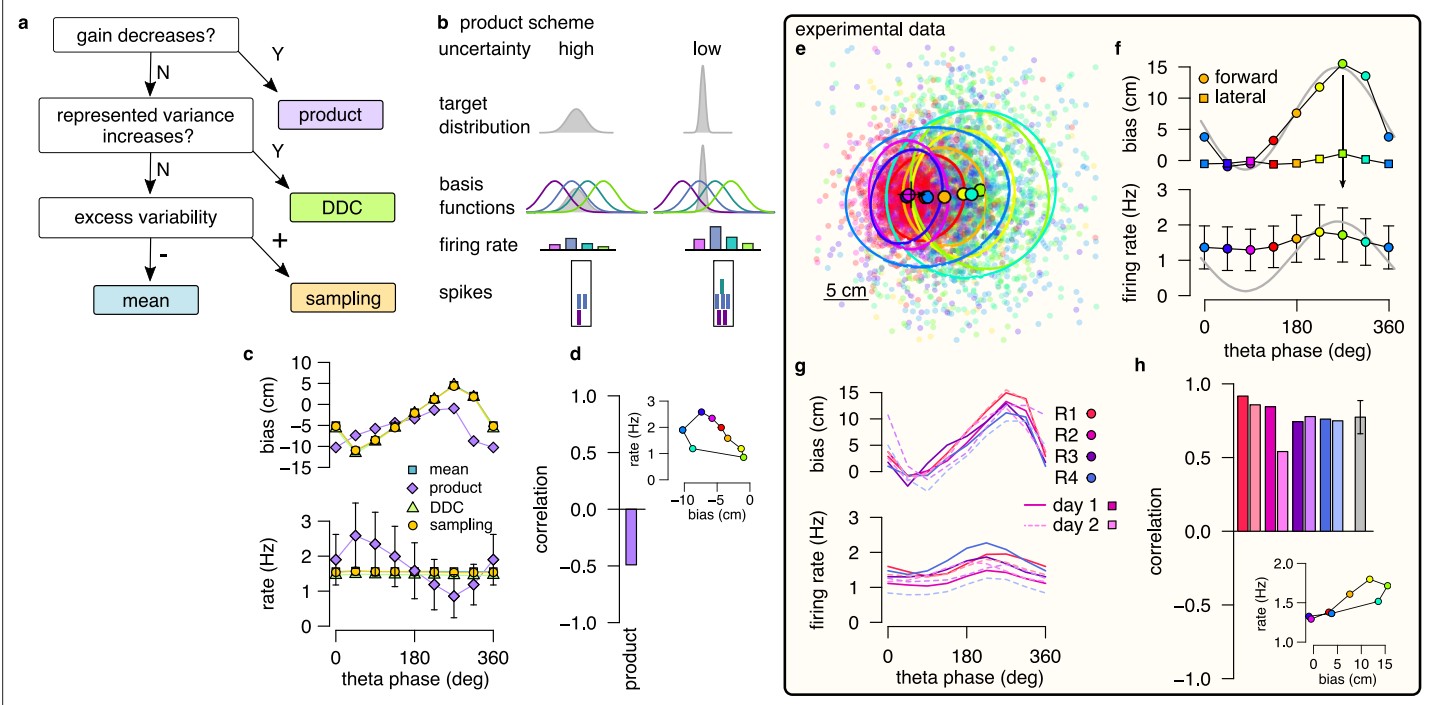

**Figure 4.** Product scheme: population gain decreases with uncertainty. (**a**) Decision-tree for identifying the encoding scheme. (**b**) Schematic of encoding a high-uncertainty (left) and a low-uncertainty (right) target distribution using the product scheme with 4 neurons. The variance is represented by the gain of the population. (**c**) Population firing rate (bottom) and forward decoding bias (top) as a function of theta phase for the four schemes in synthetic data. Only the product scheme predicts a systematic change in the firing rate. Error bars show SD across n=17990 theta cycles. (**d**) Correlation between firing rate and forward bias for the product scheme. Inset: Firing rate as a function of forward bias in the product scheme. Color code is the same as in f. (**e**) Decoded position relative to the location and motion direction of the animal (black arrow) at eight different theta phases in an example recording session. Filled circles indicate mean, ellipses indicate 50% CI of the data. (**f**) Forward decoding bias of the decoded position (top) and population firing rate (bottom) as a function of theta phase in an example recording session. Gray line in top and bottom show cosine fit to the forward decoding bias. Error bars show SD across n=9264 theta cycles. (**g**) Decoding bias (top) and firing rate (bottom) for all animals and sessions (line type). (**h**) Correlation between firing rate and forward bias for all recorded sessions. Gray bar: mean and SD across the eight sessions. Inset: Firing rate as a function of forward decoding bias in an example session. Bias in (e-g) was calculated using the 5% highest spike count theta cycles. Population firing rate plots show average over all cycles. In this figure, decoding was performed in 120° bins with 45° shifts.

The online version of this article includes the following figure supplement(s) for figure 4:

**Figure supplement 1.** Population gain as a hallmark of the product representation.

**Figure supplement 2.** Theta sequence bias and variability in all recording sessions.

**Figure supplement 3.** No evidence for encoding parameters of distributions via product or DDC schemes in linear track data.

predicted a systematic, ~threefold modulation of the firing rate within the theta cycle (*Figure 4c*, bottom). The peak of the firing rate coincided with the start of the encoded trajectory, where the uncertainty is minimal and the correlation between the firing rate and the forward bias was negative (*Figure 4d*). The three other coding schemes did not constrain the firing rate of the population to represent probabilistic quantities, and thus the firing rate was independent of the theta phase or the encoded uncertainty.

After demonstrating the specificity of uncertainty-related gain modulation to the product scheme, we returned to the experimental dataset and applied the same analysis to neuronal activity recorded from freely navigating rats. We first decoded the spikes during theta oscillation falling in eight overlapping 120° window and aligned the decoded locations relative to the animals' position and motion direction. We confirmed that the encoded location varied systematically within the theta cycle from the beginning towards the end of the theta cycle both when considering all theta cycles (*Figure 4— figure supplement 2*) or when constraining the analysis to theta cycles with the highest 5% spike count (*Figure 4e, f*; *Figure 2f, g*). Maximal population firing rate coincided with the end of the theta sequences which correspond to future positions characterized by the highest uncertainty (*Figure 4f*).

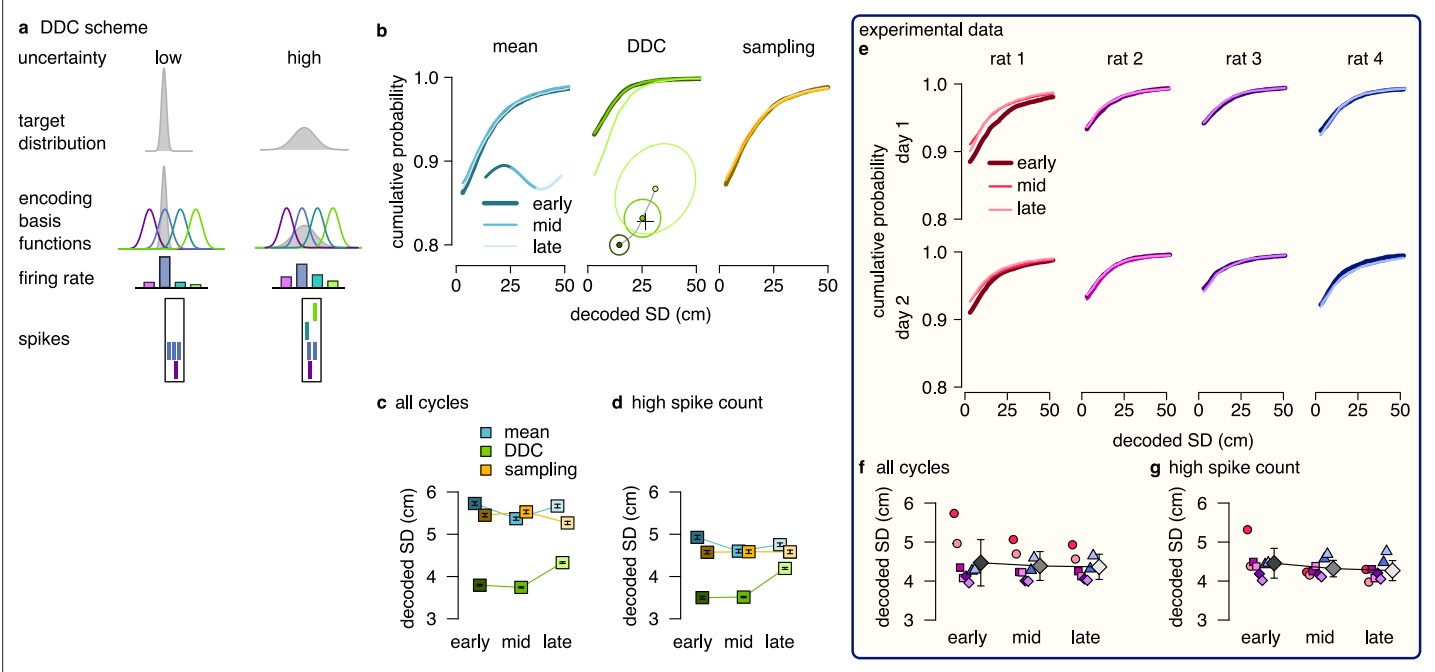

**Figure 5.** DDC scheme: diversity increases with uncertainty. (**a**) Schematic of encoding a narrow (left) and a wide (right) distribution with spike-based DDC using four neurons. Intuitively, the standard deviation (SD) is represented by the diversity of the co-active neurons. (**b**) Cumulative distribution function (CDF, accross theta cycles) of the decoded SD from spikes at different theta phases for the mean (left), DDC (middle) and sampling schemes (right) in the simulated dataset. (**c**) Mean and SE (across n=14954 theta cycles) of decoded SD as a function of theta phase for the different schemes in the simulated dataset. Only the DDC code predicts a slight, but significant increase in the decoded SD at late theta phases. (**d**) Same as panel c, calculated from theta cycles with higher than median spike count (SE across n=7214 theta cycles). (**e**) CDF of the decoded SD from spikes in early, mid and late phase of the theta cycles (across all cycles) for the analysed sessions.(**f, g**) Mean of the decoded SD for each animal from early, mid and late theta spikes using all theta cycles (**f**) or theta cycles with higher than median spikecount (**g**). Grey symbols show mean and SD across sessions. See also *Figure 5—figure supplement 2* for similar analysis using the estimated encoding basis functions instead of the empirical tuning curves for decoding and *Figure 5—figure supplement 3* for the predictions of the product scheme.

The online version of this article includes the following figure supplement(s) for figure 5:

**Figure supplement 1.** Reducing the bias of the decoded SD in the DDC scheme.

**Figure supplement 2.** No evidence for DDC code when decoding spikes using the estimated basis functions instead of the empirical tuning curves.

**Figure supplement 3.** Summary figure showing the decoded SD and EV-index for all encoding schemes.

Thus, a positive correlation emerged between the represented uncertainty and the population gain (*Figure 4h*). This result was consistent across recording sessions and animals (*Figure 4g, h*) and was also confirmed in an independent dataset where rats were running on a linear track (*Figure 4—figure supplement 3a–d*; *Grosmark et al., 2016*).

This observation is in sharp contract with the prediction of the product encoding scheme where the maximum of the firing rate should be near the beginning of the theta sequences (*Figure 4c*). The other encoding schemes are neutral about the theta modulation of the firing rate, and therefore they are all consistent with the observed small phase-modulation of the firing rate.

## Testing the DDC scheme: diversity

Next, we set out to discriminate the DDC scheme from the mean and the sampling schemes. In the DDC scheme, neuronal firing rate represents the overlap of the basis functions with the encoded distribution (*Figure 1e*; *Zemel et al., 1998*; *Vértes and Sahani, 2018*). Intuitively, in this scheme the diversity of the co-active neurons increases as the variance of the encoded distribution increases, i.e. when the encoded variance is small, only a smaller fraction of basis functions will overlap with the distribution, thus limiting the number of neurons participating in encoding. (*Figure 5a*). Conversely, a diverse set of neurons becomes co-active when encoding a distribution of high variance (*Figure 5a*). Therefore, the set of active neurons reflects the width of the encoded probability distribution. This

feature of the population code carries information about uncertainty, which can thus be decoded from the population activity (*Figure 5—figure supplement 1b*). We used a maximum likelihood decoder to estimate the first two moments (mean and SD) of the encoded distribution (Methods, *Equation 19*). Intuitively, increased uncertainty at later stages of the trajectory is expected to be reflected in a parallel increase in the decoded SD, and we propose to use this systematic increase of decoded SD as a hallmark of DDC encoding.

First, we turned to our synthetic dataset to demonstrate that systematic changes in decoded SD are specific to this scheme. In each of the three remaining coding schemes (mean, DDC and sampling), we divided the population activity to three distinct windows relative to the theta cycle (early, mid, and late). We decoded the mean and the SD of the encoded distribution of trajectories (*Figure 5b* inset) using the empirical tuning curves in each theta cycle and analysed the systematic changes in the decoded SD values from early to late theta phase (Methods). We found a systematic and significant increase in the decoded SD in the DDC scheme from early to late theta phases (one-sided, two sample Kolmogorov-Smirnov test, $P = 1.3 \times 10^{-16}$), whereas the decoded SD was independent of the theta phase for the mean and the sampling schemes (KS test, mean scheme: $P = 0.98$, sampling scheme: $P = 0.95$, *Figure 5b, c*). This result was robust against using the theta cycles with higher than median spike count for the analysis (*Figure 5d*, mean: $P = 0.99$, DDC: $P = 8.2 \times 10^{-12}$, sampling: $P = 0.93$) or against using the estimated basis functions instead of the empirical tuning curves for decoding (*Figure 5—figure supplement 2b*; Appendix 2). Thus, our analysis of synthetic data demonstrated that the decoded SD is a reliable measure to discriminate the DDC scheme from sampling or mean encoding.

After testing on synthetic data, we repeated the same analysis on the experimental dataset. We divided each theta cycle into three windows of equal spike counts (early, mid, and late) and decoded the mean and the SD of the encoded trajectories from the population activity in each window. We found that the decoded SDs had a nearly identical distribution at the three theta phases for all recording sessions (*Figure 5e*). The mean of the decoded SD did not change significantly or consistently across the recording session neither when we analysed all theta cycles (early vs. late, KS test $P > 0.62$ for all sessions, *Figure 5e, f*) nor when we constrained the analysis to the half of the cycles with higher than median spike count (KS test, $P > 0.7$ for all sessions, *Figure 5g*) or when we used the estimated encoding basis functions instead of the empirical tuning curves for decoding (*Figure 5—figure supplement 2d*). We obtained similar results in a different dataset with rats running on a linear track (*Figure 4—figure supplement 3e-i*; *Grosmark et al., 2016*). We conclude that there are no signatures of DDC encoding scheme in the hippocampal population activity during theta oscillation. Taken together with the findings on the product scheme, our results indicate that hippocampal neurons encode individual trajectories rather than entire distributions during theta sequences.

## Testing the sampling scheme: excess variability

Both the mean scheme and the sampling scheme represent individual trajectories but only the sampling scheme is capable of representing uncertainty. Therefore, a critical question concerns if the two schemes can be distinguished based on the population activity during theta sequences. Sampling-based codes are characterized by large and structured trial-to-trial neuronal variability (*Orbán et al., 2016*). Our results showed a systematic increase in the variability of the decoded location at phases of theta oscillation that correspond to later portions of the trajectory associated with higher uncertainty. This parallel increase of variability and uncertainty could be taken as evidence towards the sampling scheme. However, we demonstrated that the systematic theta phase-dependence of the neural variability and the variability of the encoded trajectories is a general feature of predictions in a dynamical model characterizing both the sampling and the mean schemes (*Figure 3*). In the sampling scheme, the cycle-to-cycle variability is further increased by the sampling variance, that is the stochasticity of the represented trajectory, such that the magnitude of excess variance is proportional to uncertainty. In order to discriminate sampling from the mean scheme we designed a measure, excess variability, that can identify the additional variance resulting from sampling. For this, we partitioned the variability of encoded trajectories such that the full magnitude of variability across theta cycles (cycle-to-cycle variability, *CCV*) is compared to the amount of variability expected from the animal's own uncertainty (trajectory encoding error, *TEE*, *Figure 6a*, Methods).

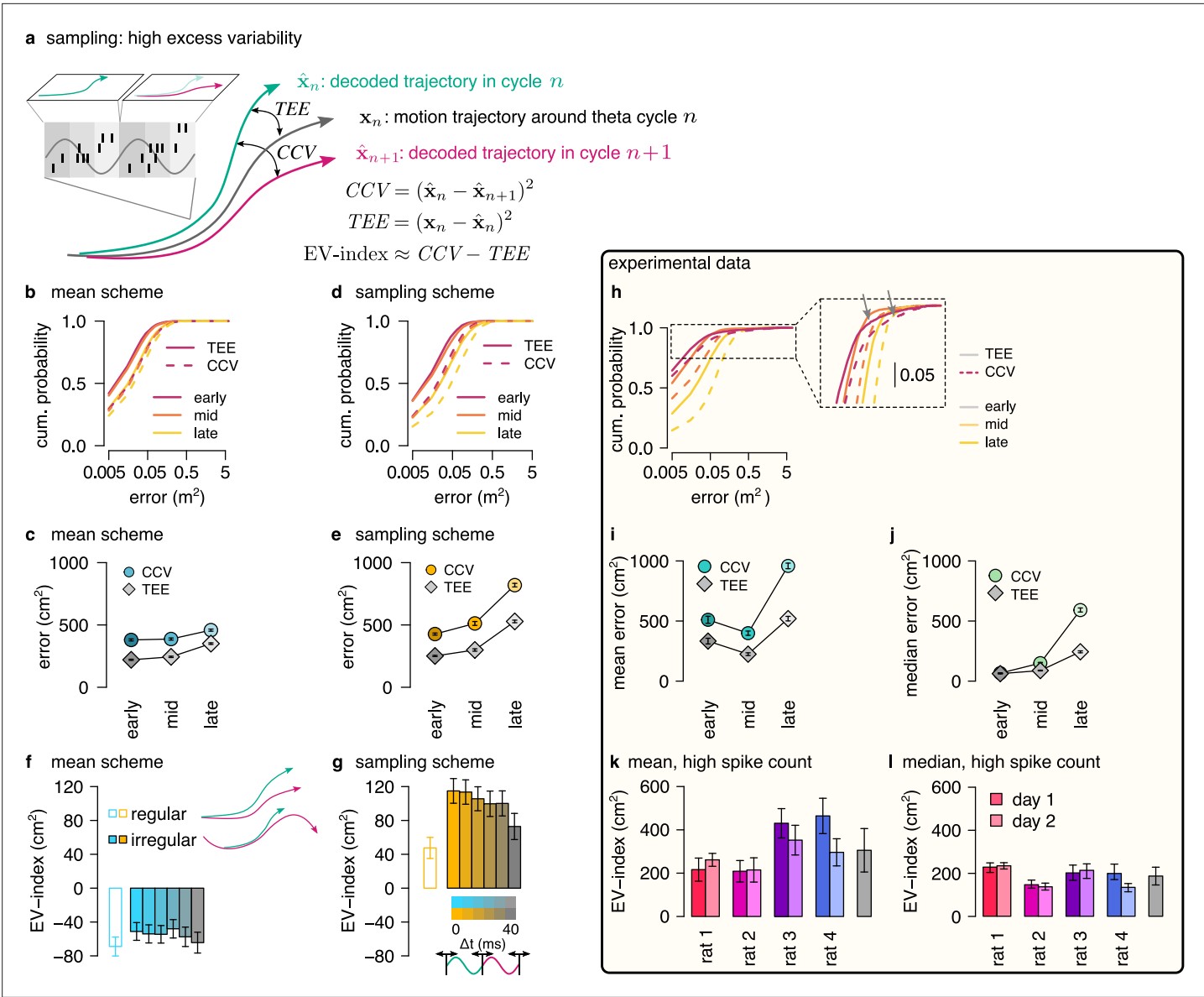

**Figure 6.** Sampling scheme: excess variability increases with uncertainty. (**a**) To discriminate sampling from mean encoding, we defined the EV-index which measures the magnitude of cycle-to-cycle variability (*CCV*) relative to the trajectory encoding error (*TEE*). (**b**) Cumulative distribution of *CCV* (dashed) and *TEE* (solid) across theta cycles for early, mid and late theta phase (colors) in the mean scheme using simulated data. Note the logarithmic x axis. (**c**) Mean *CCV* and *TEE* calculated from early, mid and late phase spikes in the mean scheme. (**d-e**) Same as (b-c) for the sampling scheme. (**f-g**) The EV-index for the mean (f) and sampling (g) schemes with simulating regular or irregular theta-trajectories (left inset) and applying various amount of jitter for segmenting the theta cycles (color code, right inset). Error bars show SE across n=5300 (regular) and n=4529 (irregular) theta cycle-pairs. (**h**) Cumulative distribution of *CCV* (dashed) and *TEE* (solid) across theta cycles for early, mid, and late theta phase (colors) for an example recording session. Note the logarithmic x axis. Arrows in inset highlight atypically large errors occurring mostly at early theta phase. (**i-j**) Mean (i) and median (j) of *CCV* and *TEE* calculated from early, mid and late phase spikes for the session shown in h. (**k-l**) EV-index calculated for all analysed sessions (color) and across session mean and SD (gray) using the mean (k) or the median (l) across theta cycles. Error bars show SE in k and 5% and 95% confidence interval in l across n=3098-6558 theta cycle-pairs. p-values are shown in *Table 3*. Here, we analysed only theta cycles with higher than median spike count. See *Figure 6—figure supplement 1* for similar analysis including all theta cycles and *Figure 5—figure supplement 3* for the EV-index calculated using the product and DDC schemes.

The online version of this article includes the following figure supplement(s) for figure 6:

**Figure supplement 1.** EV-index calculated from all theta cycles.

**Figure supplement 2.** Spatial representation and EV-index is similar across task phases.

We assessed excess variability in our synthetic dataset using either the sampling or the mean representational schemes, that is encoding either sampled trajectories or mean trajectories. Specifically, we decoded the population activity in three separate windows of the theta cycle (early, mid, and late) using a standard static Bayesian decoder and computed the difference between the decoded locations across subsequent theta cycles (cycle-to-cycle variability, *CCV*) and the difference between the decoded position and the true location of the animal (trajectory encoding error, *TEE*). As expected, both *TEE* and *CCV* increased from early to late theta phase both for mean (*Figure 6b, c*) and sampling codes (*Figure 6d, e*, Methods). Our analysis confirmed that it is the magnitude of the increase that is the most informative of the identity of the code: the increase of *CCV* is more intense within theta cycle than *TEE* in the case of sampling (*Figure 6e*) whereas the increase of the *TEE* is more intense during the theta cycle than that of *CCV* in the mean encoding scheme (*Figure 6c*, Methods). To evaluate this distinction in population responses, we quantified the difference between the rate of change of *CCV* and *TEE* using the *excess variability index* (EV-index). The EV-index was negative for mean (Methods, *Figure 6f*) and positive for sampling schemes (*Figure 6g*). To test the robustness of the EV-index against various factors influencing the cycle-to-cycle variability, we analyzed potential confounding factors. First, to compensate for the potentially large decoding errors during low spike count theta cycles, we calculated the EV-index both using all theta cycles (*Figure 6—figure supplement 1e, f*) or only theta cycles with high spike count (*Figure 6f, g*). Second, we varied randomly the speed and the length of the encoded trajectories (irregular vs. regular trajectories, *Figure 6f* inset, Methods). Third, we introduced a jitter to the boundaries of the theta cycles in order to model our uncertainty about cycle boundaries in experimental data (jitter 0–40ms, *Figure 6g* inset, Methods). We found that the EV-index was robust against these manipulations, reliably discriminating sampling-based codes from mean codes across a wide range of parameters. Thus, EV-index can distinguish sampling related excess variability from variability resulting from other sources.

We repeated the same analysis on the dataset recorded from rats exploring the 2D spatial arena. We calculated cycle-to-cycle variability and trajectory encoding error and found that typically both the error and the variability increased during theta (*Figure 6h*, inset). However, at early theta phase the distributions had high positive skewness due to a few outliers displaying extremely large errors typically at early theta phase (*Figure 6h*, arrows in the inset). The outliers could reflect an error in the identification of the start of the theta cycles, and the resulting erroneous assignment of spikes that encode a different trajectory caused increased error in the estimation of the starting position of the trajectory. To mitigate this effect, we calculated the EV-index using both the mean and the median across all theta cycles (*Figure 6—figure supplement 1j, k*) or including only high spike count cycles (*Figure 6k, l*). We found that the EV-index was consistently and significantly positive for all recording sessions. This analysis supports that the large cycle-to-cycle variability of the encoded trajectories during hippocampal theta sequences (*Gupta et al., 2012*) is consistent with random sampling from the posterior distribution of possible trajectories in a dynamic generative model.

The consistently positive EV-index across all recording sessions signified that variance in the measured responses was higher at the end of theta cycle than what would be expected from a scheme not encoding uncertainty. In fact, the magnitude of the EV-index was substantially larger when evaluated on real data than in any of our synthetic datasets, including datasets where additional structured variability was introduced through randomly changing the speed or the length of the encoded trajectories (irregular trajectories, *Figure 6g*) or through additional randomness in the cycle boundaries (jitter in *Figure 6g* cf., *Figure 6k*). A potential source for higher excess variability could be task-dependent switching between multiple coexisting maps (*Kelemen and Fenton, 2010*; *Jezek et al., 2011*). However, we found that most neurons had identical spatial tuning in the two phases of the task (random foraging versus goal directed navigation; *Figure 6—figure supplement 2a–c*) and the remaining cells typically showed relatively minor change in their spatial activity across task phases (*Figure 6—figure supplement 2*; see also *Pfeiffer, 2022*). Thus, multiple maps are not responsible for the large EV-index. Furthermore, we did not find consistent differences in the EV-index evaluated in the random foraging versus the goal directed search phase of the task (*Figure 6—figure supplement 2*), implying that specific goals during planning do not modulate the uncertainty of trajectories during theta sequences. Finally, large variability could be indicative of efficient sampling algorithms where correlations between subsequent samples are actively suppressed (*MacKay, 2003*; *Kay et al., 2020*; *Echeveste et al., 2020*). In the final section, we tested this possibility using real and simulated data.

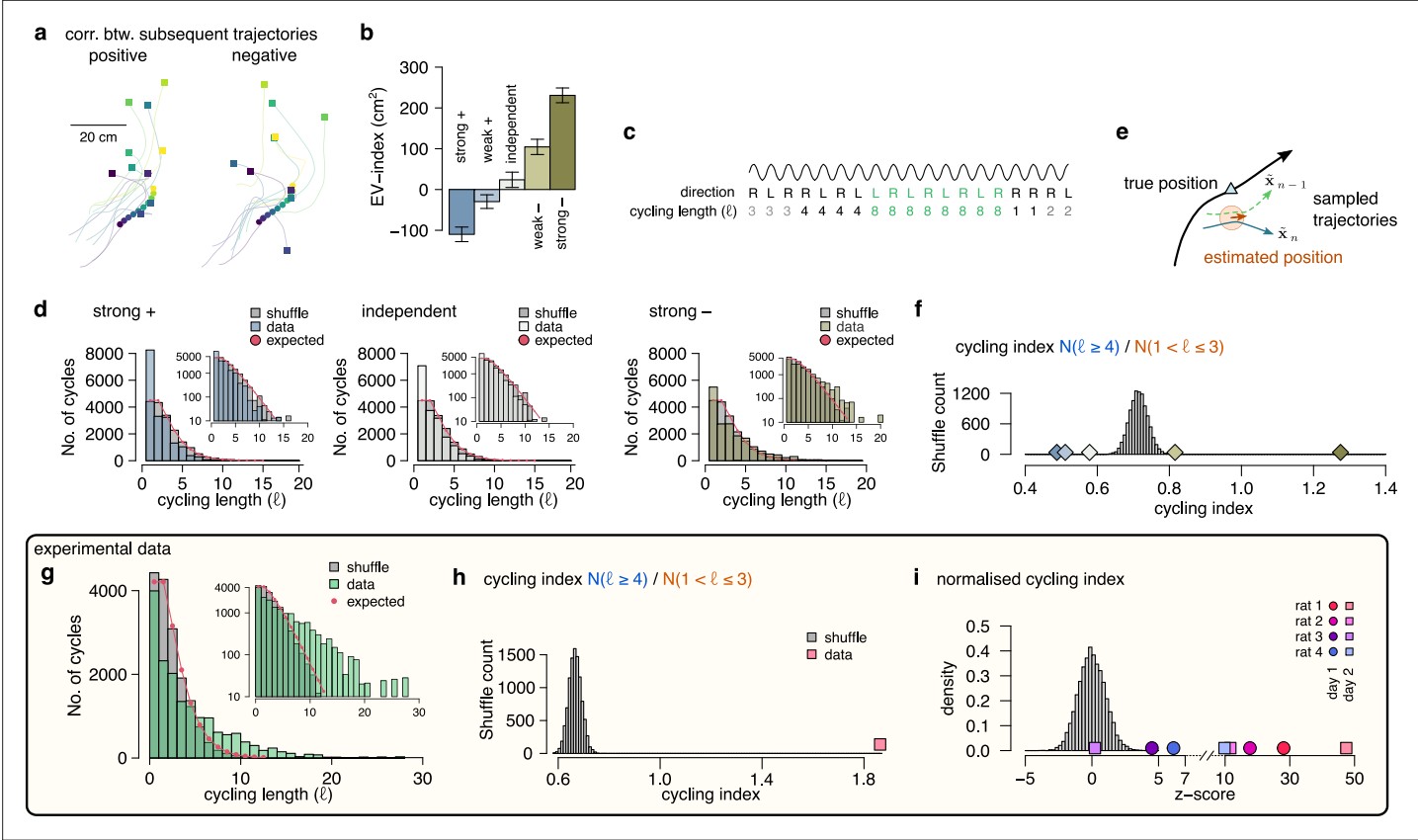

**Figure 7.** Signature of efficient sampling: generative cycling. (**a**) Examples of sampled trajectories with positive (left) and negative (right) correlation between the direction of subsequent trajectory endpoints (squares) relative to the current position (circles). (**b**) EV-index calculated with different amount of correlation between the subsequent trajectories. Error bars show SE across n=17989 theta cycle-pairs. (**c**) Schematic showing the measurement of the cycling length $\ell$. Top: theta oscillation. Middle: relative direction (L: left; R: right) of the sampled trajectories in each cycle. Bottom: cycling length ($\ell$) defined as the number of cycles with consistent alternation between left and right samples. (**d**) Histograms of the cycling length ($\ell$, see panel c) on simulated data (colors) with strong positive correlations (left), no correlations (middle) or with strong negative correlations (right). Gray shows the distribution after randomly shuffling the data across theta cycles, red shows the theoretical distribution assuming iid directions. Insets show the same distribution using logarithmic y-axes. (**e**) Schematic illustrating how difference between the true position (blue triangle) and the estimated position (brown arrowhead) can induce apparent correlations between alternating trajectories. Brown arrow represents the animal's own position and motion direction estimate with the green circle illustrating its uncertainty. Green and blue lines depict two hypothetical trajectories alternating around the animal's own position estimate (brown) but falling to the same side of its measured trajectory (black). (**f**) Cycling index ($N(\ell \geq 4)/N(1 < \ell < 3)$) in simulated data with different amount of correlation between the subsequent trajectories (colors) versus shuffled data (gray, 10,000 permutations). (**g**) Histograms of the cycling length ($\ell$) in an example session (green, R1D2) versus shuffle control (gray) and the theoretical distribution assuming iid directions (red). Inset shows the same histogram on a logarithmic scale. (**h**) Cycling index for the dataset shown in g (pink symbol) versus shuffled data (gray, 10000 permutations). (**i**) z-scored cycling index for all experimental sessions (colors) versus shuffle control (gray).

The online version of this article includes the following figure supplement(s) for figure 7:

**Figure supplement 1.** Cycling index is unbiased when the encoded position is compared to the internally estimated rather than the true position.

**Figure supplement 2.** Model-free replication of the main findings of the paper.

## Signature of efficient sampling: generative cycling

The efficiency of a sampling process can be characterized by the number of samples required to cover the target probability distribution. The efficiency can be increased when subsequent samples are generated from distant parts of the distribution making the samples anti-correlated. To test the hypothesis that the magnitude of the EV-index may be indicative of the magnitude of correlation between the samples, we first generated five datasets with varying the degree of correlation between the endpoint of trajectories sampled in subsequent theta cycles (*Figure 7a*; Methods) and calculated the EV-index for all these datasets. We found that the EV-index varied consistently with the sign and the magnitude of the correlation between the sampled trajectories: The EV-index was negative

when positive correlations between the subsequent trajectories reduced the cycle-to-cycle variability (*Figure 7b*). In this case, persistent sampling biases could not be distinguished from erroneous inference (*Beck et al., 2012*) and sampling became very similar to mean scheme. Conversely, the EV-index was positive for independent or anti-correlated samples with a substantial increase in the magnitude of the EV-index for strongly anti-correlated samples (*Figure 7b*). Thus, the large EV-index observed in the data is consistent with an efficient sampling algorithm that preferentially collects diverging trajectories in subsequent theta cycles.

To more directly test the hypothesis that subsequently encoded trajectories are anti-correlated, we explored if future portions of decoded trajectories tend to visit distinct parts of the environments. If sampling is optimized to produce anti-correlated samples then future portions of trajectories are expected to alternate around the real trajectories more intensely than positively correlated trajectories. To formulate this, we calculated the duration of the consistent cycling periods (cycling length, $\ell$, *Figure 7c*; *Kay et al., 2020*). In our synthetic dataset, we found that ch:cyclingthe cycling length took the value one more frequently than in the shuffle control, indicating the absence of alternation even when the samples were strongly anti-correlated (*Figure 7d*). We identified that the source of this bias is the difference between the observed true position of the animal, and the animal's subjective location estimate (*Figure 7e*). Specifically, even when the sampled trajectories alternate around the estimated position, the alternating trajectories often fell into the same side of the real trajectory of the animal if the true and the estimated position do not coincide (*Figure 7e*). Indeed, the bias was be eliminated when we calculated the alternation ot the encoded trajectories with respect to the animal's internal location estimate (*Figure 7—figure supplement 1*). Note that this bias is not present in simpler, 1-dimensional environments when the position of the animal is constrained to a linear corridor (*Kay et al., 2020*). To compensate for this bias, we introduced the cycling index, that ignores the repeats ($\ell = 1$) and measures the prevalence of long ($\ell \geq 4$) cycling periods relative to the short alternations ($1 < \ell \leq 3$; *Figure 7f*). We validated the cycling index on synthetic data by showing that it correctly distinguishes strongly correlated and anti-correlated settings.

Finally, we analysed the cycling behavior in the experimental data. We found that, similar to our synthetic datasets, repeats ($\ell = 1$) were relatively overrepresented in the data compared to short alternations ($\ell = 2$, *Figure 7g*). However, the distribution of cycling length had a long tail and long periods of alternations ($\ell >= 4$) were frequent (*Figure 7g*, see also *Figure 7—figure supplement 1c*) resulting in a cycling index significantly larger than in shuffle control (*Figure 7h*). To standardize the cycling index for all sessions, we z-scored it using the corresponding shuffle distribution and found that the normalized cycling index was significantly positive for 7 of the 8 recording sessions (*Figure 7i*) and not different from random in the remaining session. This finding indicates that hypothetical trajectories encoded in subsequent theta cycles tend to alternate between different directions relative to the animal's true motion trajectory, which is a signature of efficient sampling from the underlying probability distribution.

## Analysis of signatures using model-free trajectories

The core of our analysis was a generative model that we used to obtain distributions of potential trajectories in order to identify signatures of alternative coding schemes. To test if specific properties of the generative model affect these signatures, we designed an analysis in which the same signatures could be tested in a model-free manner. The original generative model directly provided a way to assess the distributions of past, current, and future potential positions using probabilistic inference in the model. In the model-free version, we used only the recorded motion trajectories to construct an empirical distribution of hypothetical trajectories at each point in the motion trajectory of the animal. Specifically, for a particular target point on a recorded motion trajectory, we sampled multiple real trajectory segments with similar starting speed and aligned the start of these trajectories with the start of the target trajectory by shifting and rotating the sampled segments (*Figure 7—figure supplement 2a*). This alignment resulted in a bouquet of trajectories for each actual point along the motion trajectory that outlined an empirical distribution of hypothetical future positions and speeds (*Figure 7—figure supplement 2a*).

We used the empirical, model-free distributions of motion trajectories to synthesize spiking data using the four different coding schemes. We then showed that the forward bias and the spread of the decoded locations changed systematically within the theta cycle (*Figure 7—figure supplement 2b,*

*c*) indicating that theta sequences are similar in this model-free dataset to real data (*Figure 1d, f and g*). Next, we repeated identical analyses that were performed with the original, model-based synthetic trajectories. Our analyses demonstrated that the signatures we introduced to test alternative coding schemes are robust against changes in the way distributions of planned trajectories are constructed (*Figure 7—figure supplement 2c–g*).

## Discussion

In this paper, we demonstrated that the statistical structure of the sequential neural activity during theta oscillation in the hippocampus is consistent with repeatedly performing inference and predictions in a dynamical generative model. Further, we have established a framework to directly contrast competing hypotheses about the way probability distributions can be represented in neural populations. Importantly, new measures were developed to dissociate alternative coding schemes and to identify their hallmarks in the population activity. Reliability and robustness of these measures was validated on multiple synthetic data sets that were designed to match the statistics of neuronal and behavioral activity of recorded animals. Our analysis demonstrated that the neural code in the hippocampus shows hallmarks of probabilistic planning by representing information about uncertainty associated with the encoded positions. Specifically, our analysis has shown that the hippocampal population code displays the signature of efficient sampling of possible motion trajectories but could not identify the hallmarks of alternative proposed schemes for coding uncertainty.

### Planning and dynamical models

Hippocampal activity sequences both during theta oscillation and sharp waves have been implicated in planning. During sharp waves, sequential activation of place cells can outline extended trajectories up to 10 m long (*Davidson et al., 2009*), providing a mechanism suitable for selecting the correct path leading to goal locations (*Pfeiffer and Foster, 2013*; *Mattar and Daw, 2018*; *Widloski and Foster, 2022*). During theta oscillation, sequences typically cover a few tens of centimeters (*Figure 2*; *Wikenheiser and Redish, 2015*) and are thus more suitable for surveilling the immediate consequences of the imminent actions. Monitoring future consequences of actions at multiple time scales is a general strategy for animals, humans and autonomous artificial agents alike (*Neftci and Averbeck, 2019*).

In our model, the dominant source of uncertainty represented by the hippocampal population activity was the stochasticity of the animal's forthcoming choices. Indeed, it has been observed that before the decision point in a W-maze the hippocampus also represents hypothetical trajectories not actually taken by the animal (*Redish, 2016*; *Kay et al., 2020*; *Tang et al., 2021*). Here, we generalized this observation to an open-field task and found that monitoring alternative paths is not restricted to decision points but the hippocampus constantly monitors consequences of alternative choices. Disentangling the potential contribution of other sources of uncertainty, including ambiguous sensory inputs (*Jezek et al., 2011*) or unpredictable environments to hippocampal representations requires analysing population activity during more structured experimental paradigms (*Kelemen and Fenton, 2010*; *Miller et al., 2017*).

Although representation of alternative choices in theta sequences is well supported by data, it is not clear how much the encoded trajectories are influenced by the current policy of the animal. On one hand, prospective coding, reversed sequences during backward travel or the modulation of the sequences by goal location indicates that current context influences the content and dynamics of the sequences (*Frank et al., 2000*; *Johnson and Redish, 2007*; *Cei et al., 2014*; *Wikenheiser and Redish, 2015*). On the other hand, theta sequences represent alternatives with equal probability in binary decision tasks and the content of the sequences is not predictive about the future choice of the animal (*Kay et al., 2020*; *Tang et al., 2021*). Our finding that there is no consistent difference between the EVindex in home versus away trials (*Figure 6—figure supplement 2*) is also consistent with the idea that trajectories are sampled from a wide distribution not influenced strongly by the current goal or policy of the animal. Sampling from a relatively wide proposal distribution is a general motif utilized by several sampling algorithms (*MacKay, 2003*). Brain areas beyond the hippocampus, such as the prefrontal cortex, might perform additional computations on the array of trajectories sampled during theta oscillations, including rejection of the samples not consistent with the current policy (*Tang et al., 2021*). The outcome of the sampling process can have profound effect on future behavior as selective

abolishment of theta sequences by pharmacological manipulations impairs behavioral performance (*Robbe et al., 2006*).

Compared to previous models simulating hippocampal place cell activity (*McClain et al., 2019*; *Chadwick et al., 2015*), a major novelty in our approach was that its spatial location was not assumed to be known by the animal. Instead, the animal had to estimate its own position and motion trajectory using a probabilistic generative model of the environment. Consequently, the neuronal activity in our simulations was driven by the estimated rather than the true positions. This probabilistic perspective allowed us to identify the sources of the variability of theta sequences, to define quantities (CCV, TEE and EV-index) that can discriminate between sampling and mean schemes and to recognise the origin of biases in generative cycling. Although the fine details of the model are not crucial for our results (*Figure 7—figure supplement 2*) and our specific parameter choices were motivated mainly to achieve a good match with the real motion and neural data (*Figure 3—figure supplements 2 and 3*), a probabilistic generative model was necessary for these insights and for the consistent implementation of the inference process. Alternative frameworks, such as the successor representations (*Dayan, 1993*; *Stachenfeld et al., 2017*) provide only aggregate predictions in the form of expected future state occupancy averaged across time and intermediate states instead of temporally and spatially detailed predictions on specific future states necessary for generating hypothetical trajectories (*Johnson and Redish, 2005*; *Kay et al., 2020*).

We found that excess variability in the experimental data was higher than that in the simulated data sampling independent trajectories in subsequent theta cycles. This high EV-index is consistent with preferentially selecting samples from the opposite lobes of the target distribution (*MacKay, 2003*) via generative cycling (*Kay et al., 2020*) leading to low autocorrelation of the samples. Recurrent neural networks can be trained to generate samples with rapidly decaying auto-correlation (*Echeveste et al., 2020*). Interestingly, these networks were shown to display strong oscillatory activity in the gamma band (*Echeveste et al., 2020*). Concurrent gamma and theta band activities are characteristics of hippocampus (*Colgin, 2016*), which indicates that network mechanisms underlying efficient sampling might be exploited by the hippocampal circuitry. Efficient sampling of trajectories instead of static variables could necessitate multiple interacting oscillations where individual trajectories evolve during multiple gamma cycles and alternative trajectories are sampled in subsequent theta cycles.

## Circuit mechanisms

Recent theoretical studies established that recurrent neural networks can implement complex nonlinear dynamics (*Mastrogiuseppe and Ostojic, 2018*; *Vyas et al., 2020*) including sampling from the posterior distribution of a static generative model (*Echeveste et al., 2020*). External inputs to the network can efficiently influence the trajectories emerging in the network either by changing the internal state and initiating a new sequence (*Kao et al., 2021*) or by modulating the internal dynamics influencing the transition structure between the cell assemblies or the represented spatial locations (*Mante et al., 2013*; *Stroud et al., 2018*). We speculate that the recurrent network of the CA3 subfield could serve as a neural implementation of the dynamical generative model with inputs from the entorhinal cortex providing strong contextual signals selecting the right map and conveying landmark information necessary for periodic resets at the beginning of the theta cycles.

Although little is known about the role of entorhinal inputs to the CA3 subfield during theta sequences, inputs to the CA1 subfield show functional segregation consistent with this idea. Specifically, inputs to the CA1 network from the entorhinal cortex and from the CA3 region are activated dominantly at distinct phases of the theta cycle and are associated with different bands of gamma oscillations reflecting the engagement of different local micro-circuits (*Colgin et al., 2009*; *Schomburg et al., 2014*). The entorhinal inputs are most active at early theta phases when they elicit fast gamma oscillations (*Schomburg et al., 2014*) and these inputs might contribute to the reset of the sequences (*Fernández-Ruiz et al., 2017*). Initial part of the theta cycle showing transient backward sequences (*Wang et al., 2020*) could reflect the effect of external inputs resetting the local network dynamics. Conversely, CA3 inputs to CA1 are coupled to local slow gamma rhythms preferentially occurring at later theta phases, associated with prospective coding and relatively long, temporally compressed paths (*Schomburg et al., 2014*; *Bieri et al., 2014*) potentially following the trajectory outlined by the CA3 place cells.

## Representations of uncertainty

Uncertainty representation implies that not only a best estimate of an inferred quantity is maintained in the population but properties of a full probability distribution can also be recovered from the activity. Machine learning provides two major classes of computational methods to represent probability distributions: 1, instantaneous representations which rely on a set of parameters to encode a probability distribution; or 2, sequential representations that collect samples from the distributions (*MacKay, 2003*). Accordingly, theories of neural probabilistic computations fall into these categories: the product scheme and DDC instantaneously, while sampling sequentially represents uncertainty (*Pouget et al., 2013*; *Savin and Deneve, 2014*). Our analysis did not find evidence for representing a probability distributions instantaneously during hippocampal theta sequences. Instead, our data is consistent with representing a single location at any given time where uncertainty is encoded sequentially by the variability of the represented locations across time.

Importantly, it is not possible to accurately recover the represented position of the animal from the observed spikes in any of the coding schemes: one can only estimate it with a finite precision, summarized by a posterior distribution over the possible positions. Thus, decoding noisy neural activity naturally leads to a posterior distribution. However, this does not imply that it was actually a probability distribution encoded in the population activity (*Zemel et al., 1998*; *Lange et al., 2020*). Specifically, when a scalar variable $x$ is encoded in the spiking activity $s(x)$ of neurons using an exponential family likelihood function, then the resulting code is a linear probabilistic population code (PPC; *Ma et al., 2006*). In fact, we used an exponential family likelihood function (Poisson) in our mean and sampling scheme, so these schemes belong, by definition, to the PPC family. However, the PPC should not be confused with our product scheme where a target distribution is encoded in the noisy population activity instead of a single variable (*Lange et al., 2020*).

To test the product scheme, we used the population gain as a hallmark. We found that the gain varied systematically, but the variance was not consistent with the basic statistical principle, that on average uncertainty accumulates when predicting future states. An alternative test would be to estimate both the encoding basis functions and the represented distributions as proposed by *Ma et al., 2006* but this would require the precise estimation of stimulus dependent correlations among the neurons.

We also did not find evidence for representing distributions via the DDC scheme during theta sequences. However, the lack of evidence for instantaneous representation of probability distributions in the hippocampus does not rule out that these schemes might be effectively employed by other neuronal systems (*Pouget et al., 2013*). In particular, on the behavioral time scale when averaging over many theta cycles, the sampling and DDC schemes become equivalent: when we calculate the average firing rate of a neuron that uses the sampling scheme across several theta cycles, it becomes the expectation of its associated encoding basis function under the represented distribution. This way, sampling alternative trajectories in each theta cycle can be interpreted as DDC on the behavioral time scale with all computational advantages of this coding scheme (*Vértes and Sahani, 2019*). Similarly, sampling potential future trajectories at the theta time scale naturally explains the emergence of successor representations on the behavioral time scale (*Stachenfeld et al., 2017*).

In standard sampling codes, individual neurons correspond to variables and their activity (membrane potential or firing rate) represent the value of the represented variable which is very efficient for sampling from complex, high dimensional distributions (*Fiser et al., 2010*; *Orbán et al., 2016*). Here, we take a population coding approach when individual neurons are associated with encoding basis functions and the population activity collectively encode the value of the variable (*Zemel et al., 1998*; *Savin and Deneve, 2014*). This scheme allows the hippocampal activity to efficiently encode the value of a low dimensional variable at high temporal precision.

Recurrent networks can implement parallel chains of sampling from the posterior distribution of static (*Savin and Deneve, 2014*) or dynamic (*Kutschireiter et al., 2017*) generative models. Similar to the DDC scheme, these implementations would also encode uncertainty by the increase of the diversity of the co-active neurons. Thus, our data indicates that the hippocampus avoids sampling multiple, parallel chains for representing uncertainty in dynamical models and rather multiplexes sampling in time by collecting several samples subsequently at an accelerated temporal scale.

Our analysis leveraged the systematic increase in the uncertainty of the predicted states with time in dynamical models. The advantage of this approach is that we could analyse 1000s of theta cycles,

much more than the typical number of trials in behavioral experiments where uncertainty is varied by manipulating stimulus parameters (e.g. image contrast; *Orbán et al., 2016*; *Walker et al., 2020*). Uncertainty could also be manipulated during navigation by changing the amount of available sensory information (*Zhang et al., 2014*), introducing ambiguity regarding the spatial context (*Jezek et al., 2011*) or manipulating the volatility of the environment (*Miller et al., 2017*). Our analysis predicts that the variability across theta cycles will increase systematically after all manipulations causing an increase in the uncertainty regardless of the nature of this manipulation or the shape of the environment. These experiments would also allow a more direct test of our theory by comparing changes in the neuronal activity with a behavioral readout of subjective uncertainty.

## Methods
### Theory
To study the neural signatures of the probabilistic coding schemes during hippocampal theta sequences, we developed a coherent theoretical framework which assumes that the hippocampus implements a dynamical generative model of the environment. The animal uses this model to estimate its current spatial location and predict possible consequences of its future actions. Since multiple possible positions are consistent with recent sensory inputs and multiple options are available to choose from, representing these alternative possibilities, and their respective probabilities, in the neuronal activity is beneficial for efficient computations. Within this framework, we interpreted trajectories represented by the sequential population activity during theta oscillation as inferences and predictions in the dynamical generative model.

We define three hierarchical levels for this generative process (*Figure 3—figure supplement 1a*). (1) We modeled the generation of *smooth planned trajectories* in the two-dimensional box, similar to the experimental setup, with a stochastic process. These trajectories represented the intended locations for the animal at discrete time steps. (2) We formulated the generation of *motion trajectories* via motor commands that are calculated as the difference between the planned trajectory and the position estimated from the sensory inputs. Again, this component was assumed to be stochastic due to noisy observations and motor commands. Calculating the correct motor command required the animal to update its position estimate at each time step and we assumed that the animal also maintained a representation of its inferred past and predicted future trajectory. (3) We modeled the generative process which *encodes the represented trajectories by neural activity*. Activity of a population of hippocampal neurons was assumed to be generated by either of the four different encoding schemes as described below.

We implemented this generative model to synthesize both locomotion and neural data and used it to test contrasting predictions of different encoding schemes. Importantly, the specific algorithm we used to synthesize motion trajectories and perform inference is not assumed to underlie the algorithmic steps implemented in the hippocampal network, it only provides sample trajectories from the distribution with the right summary statistics. The flexibility of this hierarchical framework enabled us to match qualitatively the experimental data both at the behavioral (*Figure 3—figure supplement 2*) and the neural (*Figure 3—figure supplement 3*) level. In the following paragraphs we will describe these levels in detail.

### Generation of smooth planned trajectory
The planned trajectory was established at a temporal resolution corresponding to the length of a theta cycle, $\Delta t = 0.1$ s, and spanned a length $T \approx 1$ s providing a set of planned positions, $\bar{\mathbf{x}}_n$ for any given theta cycle $n$ (see *Table 1* for a list of frequently used symbols). The planned trajectories were initialized from the center of a hypothetical 2 m×2 m box with a random initial velocity. Magnitude and direction of the velocity in subsequent steps were jointly sampled from their respective probability distributions. Specifically, at time step $n$ we first updated the direction of motion by adding a random two-dimensional vector of length $\mu_{\Delta v} = 3.5$ cm/s to the velocity $\bar{v}_n$. Next, we changed the speed, the magnitude of $\bar{v}_n$, according to an Ornstein-Uhlenbeck process:

$$\bar{\nu}_n = \bar{\nu}_{n-1} + (\mu_\nu - \bar{\nu}_{n-1})\frac{\Delta t}{\tau_\nu} + Q_\nu \epsilon \sqrt{\Delta t} \qquad (1)$$

**Table 1.** Summary of the symbols used in the model.

| Symbol | Meaning |
|---|---|
| $n$ | index of time step in the generative model measured as the number of theta cycles |
| $x_n$ | position at theta cycle $n$ (two-dimensional) |
| $y_n$ | sensory input (two-dimensional) |
| $u_n$ | motor command (two-dimensional) |
| $\bar{\mathbf{x}}_n$ | planned position ($2 \times T$-dimensional) |
| $\bar{x}_n$ | planned position (two-dimensional) |
| $y_{1:n}$ | past sensory input until theta cycle $n$ |
| $\mu_n$ | mean of the filtering posterior |
| $\Sigma_n$ | covariance of the filtering posterior |
| $\varphi$ | theta phase |
| $\mathbf{x}_n \equiv x_{(n-n_p):(n+n_f)}$ | trajectory of the animal around theta cycle $n$ |
| $\boldsymbol{\mu}_n(\varphi)$ | posterior mean trajectory at theta cycle $n$ |
| $\boldsymbol{\Sigma}_n(\varphi)$ | posterior variance of trajectory at theta cycle $n$ |
| $\tilde{\mathbf{x}}_n$ | trajectory sampled from $P(\mathbf{x}_n \vert y_{1:n}, u_{1:n})$ |
| $\phi_i(x)$ | encoding basis function of cell $i$ - firing rate as a function of the *encoded* position |
| $\psi_i(x)$ | empirical tuning curve of cell $i$ - firing rate as a function of the *real* position |
| $\lambda_i$ | firing rate of cell i |
| $\mathbf{s}_n(\varphi)$ | spikes recorded in theta cycle $n$ encoding trajectory $\mathbf{x}_n$ |
| $\hat{\mathbf{x}}_n(\varphi)$ | trajectory decoded from the observed spikes assuming direct encoding (*Equation 18*) |
| $\hat{\boldsymbol{\mu}}_n(\varphi)$ | estimated trajectory mean assuming DDC encoding (*Equation 19*) |
| $\hat{\boldsymbol{\Sigma}}_n(\varphi)$ | estimated trajectory variance assuming DDC encoding (*Equation 19*) |

where $\bar{\nu}_n$ denotes the speed in theta cycle $n$. We used the parameters $\mu_\nu = 16$ cm/s, $\tau_\nu = 2$ s, $Q_\nu = \sqrt{2/\tau_\nu}\,\sigma_\nu$ with $\sigma_\nu = 10$ cm/s and $\epsilon \sim \mathcal{N}(0, 1)$ when $2\,\text{cm/s} \leq \nu \leq 80\,\text{cm/s}$ and $\epsilon = 0$ otherwise. The planned trajectory was generated by discretized integration of the velocity signal:

$$\bar{x}_n = \bar{x}_{n-1} + \bar{\nu}_n\,\Delta t. \tag{2}$$

When the planned trajectory reached the boundary of the box, the trajectory was reflected from the walls by inverting the component of the velocity vector that was perpendicular to the wall. The parameters of the planned trajectory were chosen to approximate the movement of the real rats by the movement of the simulated animal (*Figure 3—figure supplement 2*).

Importantly, in any theta cycle multiple hypothetical planned trajectories could be generated by resampling the motion direction and the noise term, $\epsilon$ in *Equation 1*. Moreover, these planned trajectories can be elongated by recursively applying *Equations 1 and 2*. The planned trajectory influenced the motion of the simulated animal through the external control signal (motor command) as we describe it in the next paragraph.

## Generation of motion trajectories

We assumed that the simulated animal aims at following the planned trajectory but does not have access to its own location. Therefore the animal was assumed to infer the location from its noisy sensory inputs. To follow the planned trajectory, the simulated animal calculated motor commands to minimize the deviation between its planned and estimated locations.

To describe the transition between physical locations, $x$, we formulated a model where transitions were a result of motor commands, $u_n$. For this, we adopted the standard linear Gaussian state space model:

$$x_n = x_{n-1} + u_n + \varepsilon_u, \quad \varepsilon_u \sim \mathcal{N}(0, Q) \tag{3}$$

where $\varepsilon_u$ represented motor noise. The animal only had access to the location-dependent, but noisy sensory observations, $y_n$:

$$y_n = x_n + \varepsilon_y, \ \varepsilon_y \sim \mathcal{N}(0, R) \tag{4}$$

where $\varepsilon_y$ is the sensory noise and $Q$ and $R$ are diagonal noise covariance matrices with $Q_{ii} = 2.25 \text{ cm}^2$ and $R_{ii} = 225 \text{ cm}^2$. The small motor variance was necessary for smooth movements since motor errors accumulate across time (*Figure 3—figure supplement 2*). Conversely, large sensory variance was efficiently reduced by combining sensory information across different time steps (see below, *Equation 6*).

Since the location, $x_n$, was not observed, inference was required to calculate the motor command. This inference relied on estimates of the location in earlier theta cycles, the motor command, and the current sensory observation. The estimated location was represented by the Gaussian filtering posterior:

$$P(x_n | y_{1:n}, u_{1:n}) = \mathcal{N}(\mu_n, \Sigma_n). \tag{5}$$

This posterior is characterized by the mean estimated location $\mu_n$ and a covariance, $\Sigma_n$, which quantifies the uncertainty of the estimate. These parameters were updated in each time step (theta cycle) using the standard Kalman filter algorithm (*Murphy, 2012*):

$$\begin{aligned} \mu_n &= \mu_{n-1} + u_n + K_n(y_n - (\mu_{n-1} + u_n)) \\ \Sigma_n &= (I - K_n)(\Sigma_{n-1} + Q) \end{aligned} \tag{6}$$

where $K_n = (\Sigma_{n-1} + Q)(\Sigma_{n-1} + Q + R)^{-1}$ is the Kalman gain matrix.

The motor command, $u_n$, was calculated by low-pass filtering the deviation between the planned position, $\bar{x}_n$, and the estimated position of the animal (posterior mean, $\mu_{n-1}$):

$$u_n = (1 - \alpha)u_{n-1} + \alpha(\bar{x}_n - \mu_{n-1}) \tag{7}$$

with $\alpha = 0.25$ ensuring sufficiently smooth motion trajectories (*Figure 3—figure supplement 2*). Relationship between the planned position $\bar{x}_n$, actual position $x_n$, the sensory input $y_n$, and the estimated location $P(x_n | y_{1:n}, u_{1:n})$ is depicted in *Figure 3—figure supplement 1b*.

To make predictions about future positions, we defined the subjective trajectory of the animal, the distribution of trajectories consistent with all past observations and motor commands: $P(\mathbf{x}_n | y_{1:n}, u_{1:n})$. This subjective trajectory is associated with a particular theta cycle: since it is estimated on a cycle-by-cycle manner we use the index $n$ to distinguish trajectories at different cycles. Here $\mathbf{x}_n \equiv x_{(n-n_p):(n+n_f)}$ is a trajectory starting $n_p$ steps in the past and ending $n_f$ steps ahead in the future (*Figure 3—figure supplement 1c*, *Table 1*). We call the distribution $P(\mathbf{x}_n | y_{1:n}, u_{1:n})$ the *trajectory posterior*. We sampled trajectories from the posterior distribution by starting each trajectory from the posterior of current position (filtering posterior, *Equation 5*) and proceeded first backward, sampling from the conditional smoothing posterior, and then forward, sampling from the generative model.

To sample the past component of the trajectory ($m \leq n$), we capitalized on the following relationship:

$$\begin{aligned} P(x_{m-1} | x_m, y_{1:n}, u_{1:n}) &\propto P(x_{m-1} | y_{1:m-1}, u_{1:m-1}) P(x_m | x_{m-1}, u_m)) \tag{8} \\ &= \mathcal{N}\left(\Lambda_{m-1}(Q^{-1}(x_m - u_m) + \Sigma_{m-1}^{-1}\mu_{m-1}, \Lambda_{m-1}\right) \tag{9} \end{aligned}$$

where the first factor on the right hand side of *Equation 8* is the filtering posterior (*Equation 5*) and the second factor is defined by the generative process ( (*Equation 3*) ) and $\Lambda = (\Sigma_{m-1}^{-1} + Q^{-1})^{-1}$. We started each trajectory by sampling its first point independently from the filtering posterior (*Equation 5*) and applied (*Equation 8*) recursively to elongate the trajectory backward in time.

To generate samples in the forward direction ($m > n$) we implemented an ancestral sampling approach. First, a hypothetical planned trajectory was generated as in *Equations 1 and 2* starting from the last planned location $\bar{x}_n$. Next, we calculated the hypothetical future motor command, $u_{m+1}$

based on the difference between the next planned location $\bar{x}_{m+1}$ and current prediction for $m$, $x_m$ as in *Equation 7*. Finally, we sampled the next predicted position from the distribution

$$P(x_{m+1}|x_m, u_{m+1}) = \mathcal{N}(x_m + u_{m+1}, Q). \tag{10}$$

To elongate the trajectory further into the future we repeated this procedure multiple times.

We introduce $\boldsymbol{\mu}_n(\varphi) = \mathbb{E}[P(\mathbf{x}_n|y_{1:n}, u_{1:n})]$ to denote the average over possible trajectories. $\varphi$ indexes different parts of the trajectory and refers to the phase of the theta cycle at which the trajectory unfolds. Similarly, we also defined the covariance of the trajectories, $\boldsymbol{\Sigma}_n(\varphi)$. We used an approximate, diagonal covariance matrix and ignored the covariances between different trajectories and theta phase. In our simulations we estimated $\boldsymbol{\mu}_n(\varphi)$ and $\boldsymbol{\Sigma}_n(\varphi)$ from 100 samples both for the past and for the future part of the represented trajectories.

The motion profile of the simulated animal, including the distribution and the auto-correlation of the speed, acceleration and heading was matched to empirical data from real animals (*Figure 3—figure supplement 2*). *Figure 3—figure supplement 1c, d* illustrates the inference process in the model by showing a short segment of the true trajectory of the simulated animal centered on its location at time step $n$ as well as trajectories starting in the past and extending into the future sampled from the posterior distribution. As expected, the variance of these hypothetical trajectories $\boldsymbol{\Sigma}_n(\varphi)$ increased consistently from the past towards the future (from the beginning to the end of the theta cycle; illustrated by the increasing diameter of the ellipses in *Figure 3—figure supplement 1d*), while their mean $\boldsymbol{\mu}_n(\varphi)$ tracked accurately the true trajectory of the animal (*Figure 3—figure supplement 1c, d*). Mean trajectories and trajectories sampled from the trajectory posterior at subsequent theta cycles are compared in *Figure 3—figure supplement 1e, f* and in *Figure 3a, b*.

To change the correlation between the subsequent trajectories in *Figure 7*, we first generated 100 candidate trajectories sampled randomly from the posterior in time step (theta cycle) $n$ and calculated $\vartheta_n^i$, the direction of the endpoint of each of them relative to the current motion direction of the animal. Next, we calculated the absolute circular difference in the endpoint direction between the candidate directions and the direction of the endpoint of the trajectory in the previous theta cycle:

$$\Delta\vartheta_n^i = |\vartheta_n^i - \vartheta_{n-1}|_{\text{circ}} \tag{11}$$

Finally, we chose a single trajectory randomly, where the probability of each candidate trajectory was proportional to a sigmoid function of $\Delta\vartheta_n^i$

$$p_n^i \propto \frac{1}{1+\exp(-\gamma_\vartheta(\Delta\vartheta_n^i - \vartheta_0))}. \tag{12}$$

with slope ($\gamma_\vartheta$) and threshold ($\vartheta_0$) parameters controlling the sign and magnitude of the correlation between subsequent samples (*Table 2*).

## Model-free generation of motion trajectories

In *Figure 7—figure supplement 2* we replaced our generative model for potential motion trajectories by sampling trajectory segments from the real trajectory of an animal (R1D2). Specifically, we selected all 1.5 s long motion trajectory segments of continuous running and divided them into 10 quantiles based on the starting speed of the segments. To generate a distribution of potential motion trajectories for a given starting point, we collected 100 randomly sampled trajectory segments from the matching speed quantile as the reference trajectory such that all samples were consistent with the geometry of the environment (i.e. they did not cross the border of the arena). This set of trajectory segments were then used to evaluate the posterior mean trajectory ($\boldsymbol{\mu}_n(\varphi)$), the posterior variance of the trajectories ($\boldsymbol{\Sigma}_n(\varphi)$). Note, that the variance of the trajectory segments were zero at the beginning (since all segments were aligned to the same starting point). To avoid unrealistically high firing rates

**Table 2.** Parameters controlling the auto-correlation of the sampled trajectories.

| Parameter | Strong + | Weak + | Independent | Weak - | Strong - |
|---|---|---|---|---|---|
| $\gamma_\vartheta$(slope) | -8 | -5 | 0 | 5 | 5 |
| $\vartheta_0$(threshold) | $\pi/8$ | $\pi/4$ | $\pi/2$ | $\pi/4$ | $\pi/2$ |

in the product scheme, we added a constant 16 cm$^2$ to $\boldsymbol{\Sigma}_n(\varphi)$. We used these quantities to drive the neuronal activity (see below) in the same way as we used trajectories sampled from the posterior of the generative model.

## Encoding the represented trajectory by the firing of place cells

We assumed that in each theta cycle the sequential activity of hippocampal place cells represents the temporally compressed trajectory posterior. The encoded trajectory was assumed to start in the past at the beginning of the theta cycle and arrive to the predicted future states (locations) by the end of the theta cycle (*Skaggs et al., 1996*; *Foster and Wilson, 2007*). Each model place cell $i$ was associated with an encoding basis function, $\phi_i(x)$, mapping the *encoded position* to the firing rate of cell $i$. Each basis function was generated as the sum of $K$ Gaussian functions (subfields):

$$\phi_i(x) = \rho_0 + \sum_{k=1}^{K} \rho_{ik} \exp\left( (x - \mu_{ik})^{\mathrm{T}}(x - \mu_{ik})/\sigma_{ik}^2 \right) \tag{13}$$

with the following choice of the parameters:

- The number of subfields $K$ was sampled from a gamma distribution with parameters $\alpha = 0.57$ and $\beta = 1/0.14$ (*Rich et al., 2014*). We included only cells with at least one subfield within the arena ($K \geq 1$).
- The location of the subfields ($\mu$) was distributed uniformly within the environment.
- The radius of each subfield, $\sigma$, was sampled uniformly between 10 and 30 cm.
- The maximum firing rate of each subfield $\rho_{ik}$ was chosen uniformly on the range 5–15 Hz.
- The baseline firing rate $\rho_0$ was chosen uniformly on the range 0.1–0.25 Hz.

Examples of the encoding basis functions are shown in *Figure 5—figure supplement 1f* (top row). Since the *encoded* positions can be different than the *measured* positions, the encoding basis function $\phi_i(x)$ is not identical to the measured *tuning curve* $\psi_i(x)$, which is defined as a mapping from the *measured* position to the firing rate. The exact relationship between tuning curves and the encoding basis functions depends on the way the estimated location is encoded in the population activity. Tuning curves estimated from the synthetic position data are compared with experimentally recorded place cell tuning curves in *Figure 3—figure supplement 3*.

Importantly, in our model the activity of place cells was determined by the inferred trajectories and not the motion trajectory of the animal. The way the trajectories were encoded by the activity of place cells was different in the four encoding schemes:

1. In the *mean* encoding scheme the instantaneous firing rate of the place cells was determined by the mean of the trajectory posterior

$$\boldsymbol{\lambda}_i(\varphi) = \phi_i(\boldsymbol{\mu}(\varphi)) \tag{14}$$

That is, the cell's firing rate changed within the theta cycle according to the value of its basis function at the different points of the mean inferred trajectory.

2. In the *product* scheme (*Ma et al., 2006*) the firing rate was controlled both by the posterior mean and variance of the trajectory:

$$\boldsymbol{\lambda}_i(\varphi) = \phi_i(\boldsymbol{\mu}(\varphi)) \frac{\varsigma_0^2}{\varsigma^2(\varphi)} \tag{15}$$

where $\varsigma^2(\varphi) = \sqrt{\det \boldsymbol{\Sigma}(\varphi)}$ and $\varsigma_0 = 5$ cm. This is similar to the mean encoding model, except that the population firing rate is scaled by the inverse of the posterior variance.

3. In the *DDC* scheme (*Zemel et al., 1998*; *Vértes and Sahani, 2018*) the instantaneous firing rate of cell $i$ is the expectation of the basis function $i$ under the trajectory posterior at the encoded time point (that is, the overlap between the basis function and the posterior):

$$\boldsymbol{\lambda}_i(\varphi) = \int \phi_i(\mathbf{x}) \, P(\mathbf{x}(\varphi)|y_{1:n}, u_{1:n}) \, dx \tag{16}$$

4. In the *sampling* scheme, the encoded trajectory was sampled from the trajectory posterior, $\tilde{\mathbf{x}}(\varphi) \sim P(\mathbf{x}_n|y_{1:n}, u_{1:n})$ and the instantaneous firing rate was the function of the sampled trajectory:

$$\boldsymbol{\lambda}_i(\varphi) = \phi_i(\tilde{\mathbf{x}}(\varphi)) \tag{17}$$

In each encoding model, spikes were generated from the instantaneous firing rate $\lambda_i(\varphi)$ as an inhomogeneous Poisson process:

$$\mathbf{s}_i(\varphi) \sim \mathrm{Poisson}(\boldsymbol{\lambda}_i(\varphi))$$

## Decoding

To discriminate encoding schemes from each other we decoded position information both from experimentally recorded and from synthesized hippocampal neuronal activity. We performed decoding based on two different assumptions: First, assuming that a single position is encoded by the population activity (consistent with the mean and sampling schemes). Second, assuming that a distribution is encoded via the DDC scheme. We used static decoders to ensure that the variance of the decoder is independent of the theta phase as opposed to dynamic decoders, where the variance can be larger around the boundaries of the decoded segments.

### Single point decoding

We performed static Bayesian decoding independently in each temporal bin at different phases of the theta cycle. The estimated position at theta phase $\varphi$ is the mean of the posterior distribution calculated using Bayes rule:

$$\begin{aligned}
\hat{x}(\varphi) &= \int x\, P(x|\mathbf{s}(\varphi))\, dx \\
&= \int x\, \frac{\prod_i \mathrm{Poisson}(s_i(\varphi); \psi_i(x))\, P(x)}{P(\mathbf{s}(\varphi))}\, dx
\end{aligned} \tag{18}$$

where the prior $P(x)$ was the estimated occupancy map and we used a Poisson likelihood with spatial tuning curves $\psi_i(x)$, estimated from the smoothed (10 cm Gaussian kernel) and binned spike counts. We binned spikes either into windows of fixed duration (20ms, *Figure 2a–c*) fixed theta phase (120°, *Figure 3h* and *Figure 4*) or into three bins with equal number of spike counts within a theta cycle (*Figures 5 and 6* and everywhere else). When calculating the spread of the decoded locations (*Figure 2g*, *Figure 3h*), we controlled for the possible biases introduced by theta phase modulation of the firing rates by randomly downsampling the data to match the spike count histograms across theta phase.

### DDC decoding

The DDC decoder assumes that at each time point in the theta cycle an isotropic Gaussian distribution over the locations is encoded by the population activity via *Equation 16* and we aim at recovering the mean and the variance of the encoded distribution. To ensure that the theta phase dependence of the firing rates does not introduce a bias in the decoded variance, we divided each theta cycle into three windows (early, middle and late) with equal number of spikes. As linear decoding of DDC codes (*Vértes and Sahani, 2018*) from spikes is very inaccurate, we performed maximum likelihood decoding of the parameters of the encoded distribution. The estimated mean and the variance in bin $\varphi$ is:

$$\{\hat{\boldsymbol{\mu}}(\varphi),\, \hat{\sigma}^2(\varphi)\} = \underset{\boldsymbol{\mu},\, \sigma^2}{\arg\max} \prod_i \mathrm{Poisson}(\mathbf{s}(\varphi);\, \lambda_i(\boldsymbol{\mu}, \sigma^2)) \tag{19}$$

where

$$\lambda_i(\boldsymbol{\mu}, \sigma^2) = \int \phi_i(x)\, \mathcal{N}(x;\, \boldsymbol{\mu}, \sigma^2 \mathbf{I})\, dx. \tag{20}$$

Here, $\phi_i(x)$ is the basis function of neuron $i$ used in the encoding process (*Equation 16*). We numerically searched for the maximum likelihood parameters with constraints $\mu \in (0, 200)$ (cm) and $\sigma \in (3, 200)$ (cm) using quasi-Newton optimizer and a finite-difference approximation for the gradients.

In practice we do not have access to the encoding basis functions, $\phi_i(x)$, only to the empirical tuning curves, $\psi(x)$. Based on the synthetic data we found that the tuning curves are typically more dispersed than the basis functions since the encoded and the measured location is not identical or the

encoded distributions have nonzero variance (**Figure 5—figure supplement 1e, g**, Appendix 2). The difference between the size of the basis functions used for encoding and decoding introduces a bias in decoding the variance of the distribution (**Figure 5—figure supplement 1a, c**). To reduce this bias, we devised a non-parametric deconvolution algorithm that could estimate the encoding basis functions from the empirical tuning curves (**Figure 5—figure supplement 1d–f**, Appendix 2). Although we demonstrated on synthetic data that the bias of the decoder can be eliminated by using these estimated basis functions (**Figure 5—figure supplement 1d**), we obtained qualitatively similar decoding results with either the estimated basis functions or the empirical tuning curves. Therefore in **Figure 5** we show DDC decoding results obtained using the empirical tuning curves and show decoding with the estimated basis functions in **Figure 5—figure supplement 2** (see also Appendix 2).

## Analysis of trajectory variability

To discriminate the mean scheme from the sampling scheme we introduced the excess variability index (EV-index). The EV-index is the difference between the cycle-to-cycle variability and the trajectory encoding error and is positive for sampling and negative for the mean scheme. In this section, we provide definitions for these quantities, derive their expected value for the sampling and the mean scheme and show how the EV-index can be estimated from data.

### Cycle-to-cycle variability (CCV)

We defined cycle-to-cycle variability ($\chi$) as the difference between the trajectories decoded from the neural activity in two subsequent theta cycles (**Figure 3—figure supplement 1f**):

$$\chi(\varphi) \quad = \sum_i \left( \hat{\mathbf{x}}_n^i(\varphi) - \hat{\mathbf{x}}_{n-1}^i(\varphi) \right)^2 \tag{21}$$

where $\varphi$ is the theta phase and the index $i$ runs over the 2 dimensions. As we show in Appendix 3, for the mean encoding scheme the expected value of the cycle-to-cycle variability is the sum of two terms:

$$\mathbb{E}_{\text{mean}}\left[\chi(\varphi)\right] = 2\epsilon^2 + \zeta^2(\varphi) \tag{22}$$

where $\zeta^2 = \mathbb{E}[\sum_i(\boldsymbol{\mu}_n^i - \boldsymbol{\mu}_{n-1}^i)^2]$ is the expected change of the encoded mean trajectory between subsequent theta cycles, $\epsilon^2$ is the error of the decoder reflecting the finite number of observed neurons in the population and their stochastic spiking and the expectation runs across theta cycles. Since each theta cycle is divided into equal spike count bins, the decoder variance is independent of the theta phase $\varphi$. Conversely, $\zeta^2$ increases with $\varphi$ (**Figure 3—figure supplement 1g**) as new observations have larger effect on uncertain future predictions than on the estimated past positions.

Although our derivations use the expected value (mean across theta cycles), in practice we often found that the median is more robust to outliers and thus in **Figure 6** and **Figure 6—figure supplement 1** we also show results with median next to the mean.

When estimating the cycle-to-cycle variability for the sampling scheme, where the population activity encodes independent samples drawn from the trajectory posterior, an additional term appears:

$$\mathbb{E}_{\text{sam}}\left[\chi(\varphi)\right] = 2\epsilon^2 + \zeta^2(\varphi) + 2\sigma^2(\varphi). \tag{23}$$

Here, $\sigma^2(\varphi) = \mathbb{E}\left[\sum_i \boldsymbol{\Sigma}^i(\varphi)\right]$ is the total posterior variance, which reflects the variance coming from the uncertainty of the inference. In our synthetic dataset, we found that the trajectory change is proportional to the posterior variance:

$$\zeta^2(\varphi) = \alpha\,\sigma^2(\varphi) \tag{24}$$

with the proportionality constant $0 < \alpha < 1$ (**Figure 3—figure supplement 1g**). Using this insight, we can simplify our treatment: by substituting **Equation 24** into **Equations 22 and 23**, we can see that in both coding schemes the cycle-to-cycle variability increases with the theta phase, and the magnitude of this increase, proportional to the total posterior variance $\sigma^2(\varphi)$, can discriminate the two coding schemes. In order to obtain an independent estimate of $\sigma^2(\varphi)$, we can exploit insights obtained from synthetic data and introduce another measure, the trajectory encoding error.

## Trajectory encoding error (*TEE*)

We defined trajectory encoding error ($\gamma$) as the expected difference between the true two-dimensional trajectory of the rat, $\mathbf{x}_n$, and the trajectory decoded from the neural activity $\hat{\mathbf{x}}_n$ (*Figure 3—figure supplement 1e*):

$$\gamma(\varphi) = \sum_i \left( \mathbf{x}_n^i - \hat{\mathbf{x}}_n^i(\varphi) \right)^2 \tag{25}$$

When comparing decoded and physical trajectories, we assumed a fixed correspondence between theta phase $\varphi$ and temporal shift along the true trajectory. Specifically, for each animal we first calculated the average decoding error for early, mid and late phase spikes with respect to the true position temporally shifted along the motion trajectory (*Figure 4—figure supplement 2*). Next, we determined the temporal shift $\Delta t$ that minimized the decoding error separately for early, mid, and late theta phases and used this $\Delta t$ to calculate $\gamma$.

In the case of mean encoding the trajectory encoding error is the sum of two terms:

$$
\begin{aligned}
\mathbb{E}_{\text{mean}}\gamma(\varphi) \quad &= \mathbb{E}\left[ \sum_i \mathbf{x}_n^i - \boldsymbol{\mu}_n^i(\varphi) + \boldsymbol{\mu}_n^i(\varphi) - \hat{\mathbf{x}}_n^i(\varphi)^2 \right] \\
&= \boldsymbol{\sigma}^2(\varphi) + \epsilon^2
\end{aligned}
\tag{26}
$$

where we used the fact that the encoded trajectory is the mean $\mu_n(\varphi)$, and if the model of the animal is consistent, then the expected difference between the posterior mean and the true location equals the variance of the posterior, $\mathbb{E}\left[ \sum_i (\mathbf{x}_n^i - \boldsymbol{\mu}_n^i)^2 \right] = \boldsymbol{\sigma}^2$.

In the case of sampling the encoded trajectory is $\tilde{\mathbf{x}}_n$ and the trajectory encoding error is increased by the difference between the mean and the sampled trajectory:

$$
\begin{aligned}
\mathbb{E}_{\text{sam}}\left[ \gamma(\varphi) \right] \quad &= \mathbb{E}\left[ \sum_i \left( \mathbf{x}_n^i - \boldsymbol{\mu}_n^i(\varphi) + \boldsymbol{\mu}_n^i(\varphi) - \hat{\mathbf{x}}_n^i(\varphi) \right)^2 \right] \\
&= \boldsymbol{\sigma}^2(\varphi) + \mathbb{E}\left[ \sum_i \left( \boldsymbol{\mu}_n^i(\varphi) - \tilde{\mathbf{x}}_n^i(\varphi) + \tilde{\mathbf{x}}_n^i(\varphi) - \hat{\mathbf{x}}_n^i(\varphi) \right)^2 \right] \\
&= 2\boldsymbol{\sigma}^2(\varphi) + \epsilon^2
\end{aligned}
\tag{27}
$$

## Decoding error

To directly compare cycle-to-cycle variability with trajectory encoding error we have to estimate the decoding error, $\epsilon^2$. A lower bound to the decoding error is given by the Cramér-Rao bound (*Dayan and Abbott, 2001*), but in our simulations the actual decoding error was slightly larger than this bound. Underestimating the decoding error would bias the excess variability towards more positive values (see below), which we wanted to avoid as it would provide false evidence for the sampling scheme.

To obtain a reasonable upper bound instead, we note, that both $\chi$ and $\gamma$ were evaluated in three different phases of the theta cycles: early ($\varphi_1$), mid ($\varphi_2$) and late ($\varphi_3$). At early theta phases when encoding past positions the posterior variance $\sigma^2$ is small and thus both $\chi$ and $\gamma$ are dominated by $\epsilon^2$. Thus, we estimated the decoding error from the measured cycle-to-cycle variability and trajectory encoding error at early theta phase.

Furthermore, to compare cycle-to-cycle variability with trajectory encoding error we defined the compensated cycle-to-cycle variability and trajectory encoding error by subtracting the estimated decoding error:

$$\chi'(\varphi) = \chi(\varphi) - 2\epsilon^2 \quad \approx \chi(\varphi) - \chi(\varphi_1) \tag{28}$$

$$\gamma'(\varphi) = \gamma(\varphi) - \epsilon^2 \quad \approx \gamma(\varphi) - \gamma(\varphi_1) \tag{29}$$

## Excess variability

We define the excess variability as the difference between $\chi'$ and $\gamma'$:

$$\xi(\varphi) \quad = \chi'(\varphi) - \gamma'(\varphi) \tag{30}$$

**Table 3.** p-values associated with *Figure 6*.
p-values for panels f,g and k were calculated using a one sample t-test. p-values for panel l were
estimated by bootstrapping.

| Panel: f | | | | | | | |
| --- | --- | --- | --- | --- | --- | --- | --- |
| regular | jitter: 0 | 5 | 10 | 20 | 30 | 40 | |
| 8.9e-10 | 1e-06 | 4.4e-07 | 1.5e-07 | 1e-05 | 5.7e-07 | 1.8e-07 | |
| panel: g | | | | | | | |
| regular | jitter: 0 | 5 | 10 | 20 | 30 | 40 | |
| 0.0001 | 4e-15 | 3.9e-15 | 1e-13 | 3.4e-11 | 1.5e-11 | 2.4e-06 | |
| panel: k | | | | | | | |
| rat1 day1 | rat1 day2 | rat2 day1 | rat2 day2 | rat3 day1 | rat3 day2 | rat4 day1 | rat4 day2 |
| 5e-05 | 2.5e-18 | 2.5e-05 | 0.0001 | 2.1e-10 | 2.8e-07 | 1.4e-08 | 2.4e-06 |
| panel: l | | | | | | | |
| rat1 day1 | rat1 day2 | rat2 day1 | rat2 day2 | rat3 day1 | rat3 day2 | rat4 day1 | rat4 day2 |
| <0.001 | <0.001 | <0.001 | <0.001 | <0.001 | <0.001 | <0.001 | <0.001 |

Substituting *Equations 22 and 23*, *Equations 26 and 27* and *Equations 28 and 29* to *Equation 30* we can obtain the expectation of the excess variability in the sampling and the mean encoding scheme:

$$\xi_{\text{mean}}(\varphi) = (\alpha - 1)\,\sigma^2(\varphi) < 0 \tag{31}$$

$$\xi_{\text{sam}}(\varphi) = \alpha\,\sigma^2(\varphi) > 0 \tag{32}$$

The excess variability is positive for sampling and negative for mean encoding. As $\sigma^2$ is expected to increase within a theta cycle, the excess variability is most distinctive at late theta phases. Therefore, throughout the paper we defined the *EV-index* as the excess variability at late theta phases:

$$\begin{aligned} \xi(\varphi_3) \quad &= \chi'(\varphi_3) - \gamma'(\varphi_3) \\ &= \chi(\varphi_3) - \chi(\varphi_1) - \gamma(\varphi_3) + \gamma(\varphi_1) \end{aligned} \tag{33}$$

Importantly, all terms in *Equation 33* can be measured directly from the data.

We calculated the EV-index either using all theta cycles (*Figure 6—figure supplement 1*) or using theta cycles with high spike counts in order to mitigate the effect of large decoding error (*Figure 6*). To reduce the effect of outliers, we also reported the median of the EV-index in *Figure 6l and k*. Error bars on the EV-index (*Figure 6f–g and k*, *Figure 6—figure supplement 1e-f, j*) reflect the standard error across the theta cycles. When showing the median across the theta cycles (*Figure 6l*, *Figure 6—figure supplement 1k*), the error bars indicate the 5% and 95% confidence intervals estimated by bootstrapping. Specifically, we obtained 1000 pseudo-datasets by discarding randomly selected half of the theta cycles and calculating the EV-index of the remaining data. The statistical significance of the EV-index was tested using one sample t-tests (*Figure 6f, g and k*; *Figure 6—figure supplement 1e, f and j*) or bootstrapping (*Figure 6l* and *Figure 6—figure supplement 1k*). The resulting p-values are shown in *Tables 3 and 4*.

## Data analysis

### Processing experimental data

To test the predictions of the theory, we analysed the dataset recorded by *Pfeiffer and Foster, 2013*. In short, rats were required to collect food reward from one of the 36 uniformly distributed food wells alternating between random forging and spatial memory task. The rat's position and head direction were determined via two distinctly colored, head-mounted LEDs recorded by an overhead video camera and digitized at 30 Hz. Neural activity was recorded by 40 independently adjustable tetrodes

**Table 4.** p-Values associated with *Figure 6—figure supplement 1*.
p-Values for panels e,f and j were calculated using a one sample t-test. p-Values for panel k were estimated by bootstrapping.

| Panel: e | | | | | | | |
|---|---|---|---|---|---|---|---|
| regular | jitter: 0 | 5 | 10 | 20 | 30 | 40 | |
| 1.8e-05 | 1.4e-05 | 9e-05 | 4.8e-05 | 0.0016 | 0.0025 | 0.0006 | |

| panel: f | | | | | | | |
|---|---|---|---|---|---|---|---|
| regular | jitter: 0 | 5 | 10 | 20 | 30 | 40 | |
| 0.01 | 4e-05 | 7e-05 | 0.0007 | 0.0006 | 0.014 | 0.16 | |

| panel: j | | | | | | | |
|---|---|---|---|---|---|---|---|
| rat1 day1 | rat1 day2 | rat2 day1 | rat2 day2 | rat3 day1 | rat3 day2 | rat4 day1 | rat4 day2 |
| 0.0018 | 4.5e-09 | 6e-08 | 4.9e-07 | 3.7e-19 | 4e-11 | 3.3e-25 | 2.4e-07 |

| panel: k | | | | | | | |
|---|---|---|---|---|---|---|---|
| rat1 day1 | rat1 day2 | rat2 day1 | rat2 day2 | rat3 day1 | rat3 day2 | rat4 day1 | rat4 day2 |
| <0.001 | <0.001 | <0.001 | <0.001 | <0.001 | <0.001 | <0.001 | <0.001 |

targeting the left and the right hippocampi. Local field potential (LFP) was recorded on one representative electrode, digitally filtered between 0.1 and 500 Hz and recorded at 3,255 Hz. Individual units were identified by manual clustering based on spike waveform peak amplitudes based on the signals digitalized at 32,556 Hz as in *Pfeiffer and Foster, 2013*.

The raw position signal was filtered with a 250ms Gaussian kernel and instantaneous speed and motion direction was calculated from the smooth position signal. We restricted the analysis to run periods with $v > 5$ cm/s for at least 1 s separated by stops with duration of $\delta t > 0.2$ s. In this study we included only putative excitatory neurons on the basis of spike width and mean firing rate (*Pfeiffer and Foster, 2013*) with at least 200 spikes during the analysed run epochs. Position was binned (5 cm) and spatial occupancy map was calculated as the smoothed (10 cm Gaussian kernel) histogram of the time spent in each bin. Position tuning curves (ratemaps, $\psi(x)$) were calculated as the smoothed (10 cm Gaussian kernel) histogram of firing activity normalized by the occupancy map and we used $\psi_{\min} = 0.1$ Hz wherever $\psi(x) < 0.1$ Hz.

Theta phase was calculated by applying Hilbert transformation on band pass filtered (4–12 Hz) LFP signal. To find the starting phase of the theta sequences in each animal, we calculated the forward bias by decoding spikes using 120° windows advanced in 45° increments (*Figure 4h*). The forward (lateral) bias of the decoder was defined as the average of the error between the decoded and the actual position of the animal parallel (perpendicular) to the motion direction of the animal. Theta start was defined as 135° after the peak of the cosine function (*Figure 4f*) fitted to the forward bias. We used two complementary strategies to avoid biases related to systemic changes in the firing rate during decoding: (1) When analysing early, mid and late theta phases, we divided each theta cycle into three periods of equal spike count. (2) When calculating decoding spread, we subsampled the data in order to match the distribution of spike counts across theta phases. The decoding spread was defined as $\det(\Sigma)^{1/4}$ where $\Sigma$ is the covariance matrix of the decoded positions.

To calculate the theta phase dependence of the place field size (*Figure 2e*) we estimated the place fields in the three theta phase (early, mid and late) using the position of the rat shifted in time with $\Delta t(\varphi)$ that minimized the median error between the decoded and temporally shifted position in that session (*Figure 2d*). Similarly, when we calculated trajectory encoding error, we compared the decoded position in each phase bin to the real position shifted in time with the same $\Delta t(\varphi)$.

To compare place fields between home and away conditions, we first estimated separately the neuronal tuning curves for the two trial types. Home trials where the rat did not reach the goal location within 100 s were excluded from the analysis and we only included neurons with average firing rate higher than 0.1 Hz. We z-scored each tuning curve using the mean and variance of the tuning curve

estimated from all trials. This step allowed us to compare changes in the tuning curves across cells (see below).

Next, we compared the normalized tuning curve changes between the home and away trials ($\delta_{H-A}$) to two different controls. First, we randomly split the data into trials of two types 100 times and estimated normalized tuning curves from the random splits. This provided a baseline measure of the tuning curve change $\delta_{\text{shuffle}}$. Second, we obtained an upper bound on the expected tuning curve changes by comparing normalized home and away tuning curves of different neurons, $\delta_{\text{indep}}$. Neurons with the average difference between the tuning curves across random splits of the data larger than the average difference across cells ($\mathbb{E}[\delta_{\text{shuffle}}] > \mathbb{E}[\delta_{\text{indep}}]$) were considered unreliable and were excluded from further analysis. *Figure 6—figure supplement 2* shows the proportion of cells in each recording session where $\delta_{H-A}$ was larger than the 95% of distribution of $\delta_{\text{shuffle}}$, the difference between random splits of the data. To quantify the magnitude of remapping between home and away trials, we calculated the remapping index:

$$\text{RI} = \left(\delta_{H-A} - \mathbb{E}[\delta_{\text{shuffle}}]\right)/\mathbb{E}[\delta_{\text{indep}}] \tag{34}$$

which is 0 when the observed difference between home and away tuning curves ($\delta_{H-A}$) equals the mean tuning curve change in the shuffle control ($\mathbb{E}[\delta_{\text{shuffle}}]$) and is 1 when the magnitude of tuning curve change is similar to the difference between the tuning curves of independent neurons ($\mathbb{E}[\delta_{\text{indep}}]$).

### Processing synthetic data

We simulated the movement of the animal using *Equations 1–7* in a $2 \times 2$ m open arena, except for *Figure 3d, e*, where we used a $2 \times 0.1$ m long linear track. We used the same procedure to analyse the synthetic data as we applied to the experimental data: we filtered the raw position signal, calculated the speed and motion direction and estimated the spatial occupancy maps and position tuning curves from the generated spikes. Importantly, we used these estimated tuning curves and not the true, noiseless encoding basis functions for decoding position from spikes in the synthetic data. In our model, each theta cycle was 100ms long (but also see an alternative variant below) and the encoded trajectory spanned 2 s centered on the current position of the animal (*regular* theta sequences).

To test the robustness of the EV-index to variations across theta cycles (*Figure 6f, g*), we added variability to the theta cycles in two different ways: First, the duration and the content of each theta cycle was varied stochastically. Specifically, for the simulations using *irregular* theta sequences in *Figure 6f, g*, we varied the duration of each theta cycle (80–160 ms) together with the total length of the encoded trajectory (1–3 s) with constraining the encoded trajectory to start in the past and end in the future. Second, a uniformly distributed random jitter (0–40 ms) was added to the true time of the theta cycle boundaries before binning the spikes according to their theta phase.

## Acknowledgements

We thank Brad E Pfeiffer and David J Foster for kindly sharing their data and Andres D Grosmark and György Buzsáki for making their data publicly available. We thank Márton Kis and Judit K Makara for useful discussions; Mihály Bányai and Judit K Makara for comments on a previous version of the manuscript. This work was supported by an NKFIH fellowship (PD-125386, FK-125324; BBU), by the Hungarian Brain Research Program (2017–1.2.1-NKP-2017–00002, GO and KTIA-NAP-12-2-201, BBU and GO), by the Artificial Intelligence National Laboratory (European Union project RRF-2.3.1-21-2022-00004, GO).

## Additional information

### Funding

| Funder | Grant reference number | Author |
| --- | --- | --- |
| National Research, Development and Innovation Fund | PD-125386 | Balazs B Ujfalussy |

| Funder | Grant reference number | Author |
|---|---|---|
| National Research, Development and Innovation Fund | FK-125324 | Balazs B Ujfalussy |
| National Brain Research Centre | 2017-1.2.1-NKP-2017-00002 | Gergő Orbán |
| National Brain Research Centre | KTIA-NAP-12-2-201 | Balazs B Ujfalussy |
| Artificial Intelligence National Laboratory | RRF-2.3.1-21-2022-00004 | Gergő Orbán |

The funders had no role in study design, data collection and interpretation, or the decision to submit the work for publication.

### Author contributions
Balazs B Ujfalussy, Conceptualization, Data curation, Formal analysis, Funding acquisition, Writing – original draft; Gergő Orbán, Conceptualization, Funding acquisition, Writing – original draft

### Author ORCIDs
Balazs B Ujfalussy http://orcid.org/0000-0002-2295-3828
Gergő Orbán http://orcid.org/0000-0002-2406-5912

### Decision letter and Author response
Decision letter https://doi.org/10.7554/eLife.74058.sa1
Author response https://doi.org/10.7554/eLife.74058.sa2

---

## Additional files

### Supplementary files
• Transparent reporting form

### Data availability
The current manuscript is a computational study, so no data have been generated for this manuscript. The (R) code and parameters for the simulations and analysis are available at https://github.com/bbuj-falussy/tSeq (copy archived at swh:1:rev:6d131540ba45e56b46f07aba1d0cc211114d0a6d).

The following previously published dataset was used:

| Author(s) | Year | Dataset title | Dataset URL | Database and Identifier |
|---|---|---|---|---|
| Grosmark AD, Long J, Buzsáki G | 2016 | Recordings from hippocampal area CA1, PRE, during and POST novel spatial learning | https://doi.org/10.6080/K0862DC5 | Collaborative Research in Computational Neuroscience, 10.6080/K0862DC5 |

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

# Appendix 1

## Product coding scheme

In a product representation, a probability distribution over a feature, such as the position $x$, is encoded by the product of the encoding basis functions, $\phi_i(x)$:

$$P(x; \mathbf{s}) = \frac{\phi_0(x)}{\eta(\mathbf{s})} \prod_i \phi_i(x)^{s_i} \tag{35}$$

where is the number of spikes fired by neuron $i$ and $\eta(\mathbf{s}) = \int \phi_0(x) \prod_i \phi_i(x)^{s_i} \, dx$ is the normalizing factor. Alternatively, by defining a set of sufficient statistics, $h_i(x) = \log \phi_i(x)$, the same distribution can be expressed in the standard, exponential form:

$$P(x; \mathbf{s}) = \frac{\phi_0(x)}{\eta(\mathbf{s})} \exp\left( \sum_i s_i \, h_i(x) \right) \tag{36}$$

where the neuronal spikes serve as the canonical or exponential parameters of the distribution (*Wainwright and Jordan, 2007*). Intuitively, increasing the number of spikes $\mathbf{s}$ in the population adds more and more terms in *Equations 35; 36* leading to an increase in the sharpness of the represented distribution. In the special case where the tuning functions are Gaussians with variances $\sigma_i^2$ and the likelihood is conditionally independent Poisson, the represented distribution is also Gaussian with a variance, $\sigma^2(\mathbf{s})$ that is inversely proportional to the total spike count (*Ma et al., 2006*):

$$\frac{1}{\sigma^2(\mathbf{s})} = \sum_i \frac{s_i}{\sigma_i^2} \tag{37}$$

Here we show numerically that this relationship still holds for more general, multimodal basis functions, similar to the empirical tuning curves found in our dataset (*Figure 2b*). We used 110 tuning curves from an example session with mean firing rate $\bar{r} > 0.1$ Hz and tuning curve peak $r_{\max} > 10$ Hz (*Figure 4—figure supplement 1a*), we normalised the tuning curves and used their logarithm as sufficient statistics to encode distributions in the form of *Equation 36*:

$$h_i(x) = \log \psi_i(x) - \log\left( \int \psi_i(x) \, dx \right) \tag{38}$$

We used isotropic 2-dimensional Gaussians with a wide range of mean (20 cm $\geq \mu \geq$ 180 cm) and variance ($4\,\text{cm}^2 \geq \sigma^2 \geq 1600\,\text{cm}^2$) as target distributions and encoded them in the spiking activity of the neurons. *Equation 36* defines how a set of observed spikes (canonical parameters) can be interpreted as encoding a probability distribution. However, this mapping is not invertible and finding a set of spikes approximating an arbitrary target distribution is computationally challenging. We solved this problem by iteratively and greedily selecting spikes to minimise the Kullback-Leibler divergence between the encoded distribution, $Q(x) \propto \exp\left( \sum_i s_i \, h_i(x) \right)$, and the Gaussian target distribution, $P(x)$. Specifically, we first selected a single neuron whose spike minimised $\text{D}_{\text{KL}}(Q, P)$, then added more spikes one by one until the additional spikes stopped decreasing $\text{D}_{\text{KL}}(Q, P)$.

We found that the approximations were reasonably accurate, especially for distributions with smaller variance (*Figure 4—figure supplement 1b*). Importantly, the average number of spikes used for encoding decreased systematically with the variance of the encoded distribution (*Figure 4—figure supplement 1c*). Moreover, we found that even for highly variable basis functions, the total spike count in the population was well approximated as a linear function of the target precision, as expected from the theory for uniform basis functions (*Figure 4—figure supplement 1d*; *Equation 37*). Therefore we used the systematic variation of the total spike count in the population (population gain) as a function of the represented uncertainty as a hallmark of the product scheme.

When encoding trajectories in the population activity we used a simplified encoding model and scaled the gain of the neurons by the instantaneous variance of the encoded distribution (*Equation 15*).

## Appendix 2

## DDC coding scheme

In the DDC coding scheme the firing rate of neuron $i$ corresponds to the expectation of its basis function under the encoded distribution (*Zemel et al., 1998*; *Vértes and Sahani, 2018*):

$$\lambda_i = \int \phi_i(x) P(x) \, dx \tag{39}$$

This encoding scheme has complementary properties to the product scheme, as it represents exponential family distributions using their mean parameters instead of their canonical, natural parameters (*Wainwright and Jordan, 2007*). *Equation 39* defines a mapping from a distribution to firing rates, but the reverse mapping, from the rates to the encoded distribution is not trivial. When the firing rates are observed then the expectation of any nonlinear function $f(\cdot)$ under the encoded distribution, including its moments, the mean and the covariance, can be calculated as a linear combination of the rates (*Vértes and Sahani, 2018*):

$$\mathbb{E}[f(x)]_{P(x)} = \sum_i \alpha_i \lambda_i \tag{40}$$

where $f(x) \approx \sum_i \alpha_i \phi_i(x)$. However, in the experimental data only the discrete spike counts are observed and the underlying firing rates are hidden. In this case *Equation 40* can be computed as an expectation under the posterior of the firing rates $P(\boldsymbol{\lambda}|\mathbf{s})$:

$$\mathbb{E}[f(x)]_{P(x)} = \int P(\boldsymbol{\lambda}|\mathbf{s}) \sum_i \alpha_i \lambda_i \, d\boldsymbol{\lambda} \tag{41}$$

with

$$P(\boldsymbol{\lambda}|\mathbf{s}) \propto P(\boldsymbol{\lambda}) \prod_i P(s_i|\lambda_i) \tag{42}$$

Here the prior over the firing rates $P(\boldsymbol{\lambda})$ (the distribution of coactivity patterns among neurons with different tuning curves or place fields) is usually assumed to be factorised for computational efficiency. However, correlations between neuronal tuning curves introduce strong dependence in the prior and thus using a factorised approximation of *Equation 41* leads to a substantial loss of information (*Ujfalussy et al., 2015*). Therefore, instead of following this Bayesian approach, we aimed at directly estimating the parameters of the distribution encoded by the spikes observed in a neuronal population using a maximum likelihood decoder (*Equation 19*).

We tested the performance of the maximum likelihood DDC decoder by encoding isotropic 2-dimensional Gaussian distributions with a wide range of mean (20 cm $\geq \mu \geq$ 180 cm) and variance (40 cm$^2 \geq \sigma^2 \geq$ 400 cm$^2$) as target in the spiking activity of a neural population using 200 encoding basis functions similar to the empirical tuning curves recorded experimentally (*Figure 6—figure supplement 1*, top). We calculated the firing rate of the neurons according to *Equation 39* and generated Poisson spike counts in $\Delta t = 20 - 100$ ms time bins. We found that on average, we could accurately estimate the encoded mean and SD of the target distributions when we used the same basis functions during encoding and decoding at least for $\Delta t \geq 50$ ms. (*Figure 5—figure supplement 1b*). However, in practice, we do not have access to the encoding basis functions, only the empirical tuning curves, substantially wider than the basis functions (*Figure 5—figure supplement 1e*). When we used the wider tuning curves for decoding, the estimate of the SD became substantially biased: the decoded SD could be on average 5–7 cm below the target SD even for $\Delta t = 100$ ms (*Figure 5—figure supplement 1e*).

To identify DDC scheme in the data it is important to be able to reduce this decoding bias and accurately estimate the SD of the encoded distribution even in small time windows. We speculated that this bias is due to the difference between the wider empirical tuning curves used for decoding and the narrower basis functions used for encoding the target distributions. This effect is caused by encoding locations other than the actual position of the animal and by encoding distributions with non-zero variance using the DDC scheme. The decoder tries to match the overlap between the tuning functions and the estimated distribution with the observed spike counts. When the tuning functions used for DDC-decoding are wider than the encoding tuning functions, the decoder will compensate this by choosing a narrower decoded distribution to achieve a similar overlap with the

decoding tuning functions (*Figure 5—figure supplement 1a*). Thus, the decoder will be biased towards lower variances, leading to systematic underestimation of the variance of the encoded distribution.

In order to reduce this bias we aimed at estimating the true encoding basis functions from the recorded neuronal activity. Note, that deriving an optimal estimator is hindered by the fact that in *Equation 39* neither the tuning functions nor the encoded distributions or the true firing rates are observed. Therefore we developed an approximate algorithm to sharpen the empirical tuning curves and reconstruct the original tuning functions. The aim was to find a flexible algorithm that preserves the relative mass of different modes of the tuning curves but sharpens all of them to a similar degree. Our method was inspired by particle filtering and the use of annealing methods in sampling (*Murray, 2007*): we first generated a set of samples from the empirical tuning curves, and then added random noise to propagate the samples towards the desired distribution. Specifically, we used the following procedure to sharpen the tuning curves:

1. We generated $N = 10,000$ samples from the empirical tuning curve $z \sim P(z) = \frac{1}{\eta}\psi_i(z)$, where $\eta = \int \psi_i(z)\,\mathrm{d}z$ is the normalization constant to convert the tuning curve to a distribution.
2. We added a Gaussian random noise to the samples $z' = z + \epsilon$ with $\epsilon \sim \mathcal{N}(0, \sigma^2)$ where the proposal variance, $\sigma^2$ controls the degree of mixing between the different modes. We used $\sigma = 10$ cm.
3. We accepted the new sample $z'_k$ with probability $p_k = \min\left(1, \left(P(z'_k)/P(z_k)\right)^\beta\right)$ where the parameter $\beta$ controls the sign and degree of sharpening. We used $\beta = 3$.
4. We repeated steps 2. and 3. for a number of iterations, not necessarily until convergence.
5. Finally, we defined the estimated basis functions as the smoothed (5 cm Gaussian kernel) histogram of the number of samples in 5 cm spatial bins.

*Figure 5—figure supplement 1g* shows the tuning functions of a few example cells from our synthetic dataset (top row) together with the measured tuning curves (second row) and the result of the sharpening algorithm after 3 and 6 steps (3rd and 4th rows). We evaluated the sharpening algorithm by comparing the relative size of the estimated and the original tuning functions and the average difference between them. We found that 6 iterations of the algorithm eliminates overestimation bias of the tuning curve size (*Figure 5—figure supplement 1e*) and 3–6 iterations minimized the estimation error (*Figure 5—figure supplement 1f*). Using the 3-step sharpened tuning curves also improved the performance of the DDC-decoder as it substantially reduced the bias when decoding the SD of the distribution (*Figure 5—figure supplement 1d*). Therefore we repeated the DDC decoding analysis shown in *Figure 5* using the 3-step sharpened tuning curves in *Figure 5—figure supplement 2*.

Even with this correction, trial by trial decoding of the represented distributions was not possible on short time scales. However, when averaged across 1000s of theta cycles, the decoder became sufficiently sensitive to the encoded SD to identify systematic changes in the represented uncertainty. Therefore, we used the average SD decoded from the neural activity as a hallmark of DDC encoding.

# Appendix 3

## Background for the calculation of EV-index

### Cycle-to-cycle variability

Here we derive the expectation of *Equation 21* in the case of mean encoding scheme, when the population activity encodes the posterior mean trajectory $\boldsymbol{\mu}_n$:

$$
\begin{aligned}
\mathbb{E}_{\text{mean}}\left[\sum_i(\hat{\mathbf{x}}_n^i - \hat{\mathbf{x}}_{n-1}^i)^2\right] &= \mathbb{E}\left[\sum_i\left((\hat{\mathbf{x}}_n^i - \boldsymbol{\mu}_n^i) - (\hat{\mathbf{x}}_{n-1}^i - \boldsymbol{\mu}_n^i)\right)^2\right] \\
&\approx \mathbb{E}\left[\sum_i(\hat{\mathbf{x}}_n^i - \boldsymbol{\mu}_n^i)^2 + \sum_i(\hat{\mathbf{x}}_{n-1}^i - \boldsymbol{\mu}_n^i)^2\right] \\
&= \epsilon^2 + \mathbb{E}\left[\sum_i(\hat{\mathbf{x}}_{n-1}^i - \boldsymbol{\mu}_{n-1}^i + \boldsymbol{\mu}_{n-1}^i - \boldsymbol{\mu}_n^i)^2\right] \\
&\approx 2\epsilon^2 + \zeta^2
\end{aligned}
\tag{43}
$$

where $\zeta^2 = \mathbb{E}\left[\sum_i(\boldsymbol{\mu}_n^i - \boldsymbol{\mu}_{n-1}^i)^2\right]$ is the expected change of the posterior mean trajectory due to sensory inputs, and we omitted second order terms including the correlations between the decoding errors in subsequent theta cycles and the correlation between decoding error and trajectory change.

The same expectation in the case of sampling, when the encoded location $\tilde{\mathbf{x}}_n$ is sampled from the posterior:

$$
\begin{aligned}
\mathbb{E}_{\text{sam}}\left[\sum_i(\hat{\mathbf{x}}_n^i - \hat{\mathbf{x}}_{n-1}^i)^2\right] &= \mathbb{E}\left[\sum_i\left((\hat{\mathbf{x}}_n^i - \tilde{\mathbf{x}}_n^i) - (\hat{\mathbf{x}}_{n-1}^i - \tilde{\mathbf{x}}_n^i)\right)^2\right] \\
&\approx \epsilon^2 + \mathbb{E}\left[\sum_i(\hat{\mathbf{x}}_{n-1}^i - \tilde{\mathbf{x}}_{n-1}^i + \tilde{\mathbf{x}}_{n-1}^i - \tilde{\mathbf{x}}_n^i)^2\right] \\
&\approx 2\epsilon^2 + \mathbb{E}\left[\sum_i(\tilde{\mathbf{x}}_n^i - \tilde{\mathbf{x}}_{n-1}^i)^2\right] \\
&= 2\epsilon^2 + \mathbb{E}\left[\sum_i\left((\tilde{\mathbf{x}}_n^i - \boldsymbol{\mu}_n^i + \boldsymbol{\mu}_n^i) - (\tilde{\mathbf{x}}_{n-1}^i - \boldsymbol{\mu}_{n-1}^i + \boldsymbol{\mu}_{n-1}^i)\right)^2\right] \\
&\approx 2\epsilon^2 + \mathbb{E}\left[\sum_i(\tilde{\mathbf{x}}_n^i - \boldsymbol{\mu}_n^i)^2 + \sum_i(\boldsymbol{\mu}_{n-1}^i - \tilde{\mathbf{x}}_{n-1}^i)^2 + \sum_i(\boldsymbol{\mu}_n^i - \boldsymbol{\mu}_{n-1}^i)^2\right] \\
&= 2\epsilon^2 + 2\sigma^2 + \zeta^2
\end{aligned}
\tag{44}
$$

where $\sigma^2 = \mathbb{E}\left[\sum_i\boldsymbol{\Sigma}^i\right]$ measures the total variance of the posterior and we omitted all second order terms. In our numerical simulations we found that the contribution of second order terms are small in most cases, except for the *border-effect* which captures the correlation of the decoding error in two subsequent theta cycles. The relatively strong positive correlation is explained by the bias of the decoder near the borders of the arena. Importantly, the contribution of this term is independent of theta phase and thus does not influence our analysis. We also assumed that the sample auto-correlation is near zero i.e., the samples drawn in subsequent theta cycles are independent.

