## [Editor Report]

This paper will be of interest to neuroscientists interested in predictive coding and planning. It presents a novel analysis of hippocampal place cells during exploration of an open arena. It performs a comprehensive comparison of real and synthetic data to determine which encoding model best explains population activity in the hippocampus.

---

## [Decision Letter]

**Decision letter after peer review:**

Thank you for submitting your article "Sampling motion trajectories during hippocampal theta sequences" for consideration by *eLife*. Your article has been reviewed by 3 peer reviewers, and the evaluation has been overseen by a Reviewing Editor and Laura Colgin as the Senior Editor. The reviewers have opted to remain anonymous.

Essential revisions:

1) The demonstration that place cell firing samples potential future trajectories from the current location has so far only been shown in narrow armed mazes. This study, based on hippocampal recording in rats exploring 2D environments, is thus an interesting new contribution. However, there are several aspects of the analyses that should be clarified. Most importantly, it was recently shown that future sweeps of place cells alternate left/right on a T-maze (Kay et al., 2020). Grid cells may do the same in open environments (Gardner R. J., Vollan A. Z., Moser M.-B., Moser E. I. (2019) Soc. Neurosci. Abstr. 604.13/AA9). This study would thus benefit from bringing its original framing more up to date and from analyzing whether consecutive sweeps are anticorrelated.

2) Then, it is possible that the sampling of future trajectories corresponds to the representation of multiple coexisting "maps" (e.g. Jackson and Redish Hippocampus 2007, Kelemen and Fenton PLoS Biol 2010), especially since the animals alternated between two different behavioural strategies, namely the search of the "away" location and going back to the "home" location. This "flickering" between maps could potentially explain the over-dispersion of the data, resulting in variance that could be interpreted as sampling of possible future trajectories. Additional analyses could clarify this potential bias.

3) The generative model seems to be a key aspect of the study. A lot of work has clearly been done to model as accurately as possible an animal's movement. However, it also raises the question of how critical this is for the whole analysis. Are the predictions somehow different with a simpler model, for example one that would be generated as a successor representation? It is possible that generating naturally looking trajectories this way is not possible, but it would be interesting to at least discuss why all components of the generative model are strictly necessary. In other words, to discuss why the same predictions cannot be done with a simpler model.

4) It is unclear how the diversity prediction of the DDC model was tested. Specifically, how the variance in panel 5b was computed?

5) Not all "signatures" were shown for all models. The study would benefit from a summary figure or table showing how each model's prediction compared to the data for each of the signatures.

*Reviewer #1 (Recommendations for the authors):*

I think this paper should be presented from the point of view that theta sweeps are already thought to represent specific potential trajectories extending from the current location, from extensive work on mazes. However, in open field data (where trajectories are not constrained to specific routes), although there are similar reports concerning replay in open fields and theta sweeps in entorhinal cortex, it is still possible that place cells actually represent the distribution of possible future trajectories, and you show that this is not the case, providing some of the first evidence that theta sweeps in open fields encode specific trajectories.

I did not find it helpful that the paper is framed as evaluating different possible neural representations of uncertainty, and I don't think the rejection of the product or DDC schemes for doing this necessarily tells us much about whether or not they are used in situations where the brain might in fact represent a probability distribution.

*Reviewer #3 (Recommendations for the authors):*

Please provide more details how the increased diversity prediction of the DDC was tested. How is the diversity decoded from the population? What do the cumulative probabilities on Figure 5 subplot b show exactly and why and how can they show the increased diversity? The authors may want to provide some intuition/more explanation about what the decoded SD reflects in Figure 5 b, d and why it reflects the diversity and not the uncertainty which would increase in the late cycles for the rest of the models too.

We assume all analyses ("signatures") were applied to all models but not all of the results are shown. We appreciate that the step-by-step approach eliminating one model at a time makes for a simpler story but a figure with all results would seem to be very informative: e.g. are the other signatures that contradict the product scheme?

Presentation: we did not understand the cumulative probability panels and how they are related to the text. E.g. lines 249-252 and Figure 5b, same for Figure 6. Maybe it would be more intuitive to show the pdf instead of the cdf? But even then, the x-axis and general interpretation remains unclear to us.

Line 333: How do the results suggest efficient planning in the hippocampus? It suggests probabilistic, i.e. close to statistically optimal computations, but the authors should provide more details why they think it is also efficient.

[Editors’ note: further revisions were suggested prior to acceptance, as described below.]

Thank you for resubmitting your work entitled "Sampling motion trajectories during hippocampal theta sequences" for further consideration by *eLife*. Your revised article has been evaluated by Laura Colgin (Senior Editor) and a Reviewing Editor.

The manuscript has been improved but reviewer #2 has one remaining issue that needs to be addressed, as outlined below.

*Reviewer #1 (Recommendations for the authors):*

The authors have answered my main concerns

*Reviewer #2 (Recommendations for the authors):*

In this revision, the authors have made a number of improvements to what was already a systematic, rigorous examination of an important issue. The control analysis ruling out excess variance is due to different place maps across navigation-to-goal and random foraging further enhances confidence in the results, and is additionally supported by a similar analysis in Brad Pfeiffer's paper that just came out (PMID: 35396328).

My one remaining hesitation with this paper is the one I brought up in my previous review but perhaps didn't explain clearly, so I will try again. My understanding of the core logic of the paper is that the authors first establish what the signatures are of the coding schemes they wish to distinguish, by implementing these schemes in a series of models that generate synthetic spiking data. Then, they test to what extent those signatures exist in real data, and draw inferences from comparison with the simulated results.

I think this is a powerful approach and am completely on board with it. However, it does raise an overall question of how robust the simulation results are: what components of the simulation are necessary and sufficient for the results? How sensitive are the results to specific parameter choices? I appreciate and agree with the authors' argument that the Kalman filter is an appropriate and relatively minimal way to model the animal's uncertainty about its own location. But it is not obvious to me what modeling the animal's uncertainty about its position contributes to the results in the first place. Would you get the same simulated signatures if all you had was a probability distribution of expected future trajectories given true current location, and then applied the various coding schemes (encoding MAP trajectory, sampling, etc)?

Hence, my suggestion of using the SR to obtain such trajectory distributions from the animal's behavioral data, but really any approach that estimates these distributions from the data would help address my concern. This feels important, because comparison between these different generative models would give a sense of what data sets/behavioral tasks/task conditions should show the predicted signatures. Ultimately we don't want to just understand what happens in the Pfeiffer and Foster 2013 data set the authors analyze, but have some idea of what to expect say, in the dark with high uncertainty about current location, or on armed mazes where possible futures are highly constrained.

I realize that the paper is already extensive and thorough, but unless I'm off base with this intuition or misunderstand the authors' argument, it could actually simplify their paper if the same results obtain with a simpler way of generating probability distributions over trajectories. If they don't obtain, then this is important to point out, because of the resulting prediction that theta sequences ought to have different spiking statistics in high vs low uncertainty-about-current-position conditions.

*Reviewer #3 (Recommendations for the authors):*

The authors have addressed all my concerns. They have added an interesting new analysis as the result of the reviewer suggestions. Congratulations to the authors for an impressive piece of work.

In the final version, I'd like to encourage the authors to better explain the cause of the bias towards similar directions described in line 404. I didn't understand it. Please elaborate.

---

## [Author Response]

Essential revisions:1) The demonstration that place cell firing samples potential future trajectories from the current location has so far only been shown in narrow armed mazes. This study, based on hippocampal recording in rats exploring 2D environments, is thus an interesting new contribution. However, there are several aspects of the analyses that should be clarified. Most importantly, it was recently shown that future sweeps of place cells alternate left/right on a T-maze (Kay et al., 2020). Grid cells may do the same in open environments (Gardner R. J., Vollan A. Z., Moser M.-B., Moser E. I. (2019) Soc. Neurosci. Abstr. 604.13/AA9). This study would thus benefit from bringing its original framing more up to date and from analyzing whether consecutive sweeps are anticorrelated.

We thank the reviewers for encouraging us to analyse the autocorrelation structure of the theta sequences. We performed the suggested analysis and our results confirm and extend previous findings that trajectories encoded during theta sequences are indeed anti-correlated in 2-dimensional environments. Since this result also underscores one of the main messages of the paper, i.e., that the brain is able to perform probabilistic computations efficiently: correlated samples require a higher number of samples to represent a probability distribution, and therefore decorrelation can contribute to a code that requires less samples and consequently less time to perform more accurate inference. We present these results on a new main figure (Figure 7) and associated section in the main text (Signature of efficient sampling:generative cycling, line ~382-425). We also added a corresponding supplemenal figure (Figure 7 figure supplement 1).

2) Then, it is possible that the sampling of future trajectories corresponds to the representation of multiple coexisting "maps" (e.g. Jackson and Redish Hippocampus 2007, Kelemen and Fenton PLoS Biol 2010), especially since the animals alternated between two different behavioural strategies, namely the search of the "away" location and going back to the "home" location. This "flickering" between maps could potentially explain the over-dispersion of the data, resulting in variance that could be interpreted as sampling of possible future trajectories. Additional analyses could clarify this potential bias.

Thank you for raising this issue. We performed the analysis to assess the differences in the behaviour and the spatial tuning of the recorded neurons between the two task phases (home versus away runs). Although we found significant differences in spatial tuning of a minority of the cells, these differences were relatively small and thus we do not believe that this flickering could substantially contribute to the overdispersion of the data. We included this analysis as a supplemental figure (Figure 6 figure supplement 2)

3) The generative model seems to be a key aspect of the study. A lot of work has clearly been done to model as accurately as possible an animal's movement. However, it also raises the question of how critical this is for the whole analysis. Are the predictions somehow different with a simpler model, for example one that would be generated as a successor representation? It is possible that generating naturally looking trajectories this way is not possible, but it would be interesting to at least discuss why all components of the generative model are strictly necessary. In other words, to discuss why the same predictions cannot be done with a simpler model.

The generality of the signatures follows from the fact that we derived them from the fundamental properties of the encoding schemes, not the generative model. The sole criterion for the generative model was to produce synthetic trajectories that match experimental trajectories well (Figure 3, figure supplement 2) and to enable simple inference in the model (Figure 3, figure supplement 1). We tested the robustness of signatures using both idealized test data (Figure 4, figure supplement 1c-d, Figure 5, figure supplement 1) and our simulated hippocampal model (Figure 4c, Figure 5B-c, Fig6b-g). We added a paragraph discussing the choice of the generative model and clarified the relationship between our model and the successor representation framework in the discussion (lines 486-490).

4) It is unclear how the diversity prediction of the DDC model was tested. Specifically, how the variance in panel 5b was computed?

Thank you for raising the issue of clarity of presentation. We extended the description of DDC decoding in the main text (lines 268-273) and added a reference to the Methods section where the mathematical details are provided (Equations 17-18).

Briefly, In the DDC scheme we assumed that the hippocampal population activity at each time point encodes the probability distribution using the overlap between the neuronal basis functions and the target distribution. Just as we can use a static Bayesian decoder to decode the mean of the encoded location when a single position is encoded (as in the mean encoding scheme), we can also decode higher moments (e.g., variance) from a set of spikes when a full distribution is represented in the population activity (as in the product or the DDC schemes). In this case, we made a further assumption, that an isotropic Gaussian distribution was encoded, and used a maximum likelihood estimation to infer the three parameters of the encoded distribution (x and y position as well as the standard deviation).

5) Not all "signatures" were shown for all models. The study would benefit from a summary figure or table showing how each model's prediction compared to the data for each of the signatures.

We added a novel supplemental figure (Figure 5—figure supplement 3.) showing all signatures for all Models.

Reviewer #1 (Recommendations for the authors):I think this paper should be presented from the point of view that theta sweeps are already thought to represent specific potential trajectories extending from the current location, from extensive work on mazes. However, in open field data (where trajectories are not constrained to specific routes), although there are similar reports concerning replay in open fields and theta sweeps in entorhinal cortex, it is still possible that place cells actually represent the distribution of possible future trajectories, and you show that this is not the case, providing some of the first evidence that theta sweeps in open fields encode specific trajectories.

We thank the reviewer for this suggestion. We added a new paragraph (lines 74-88) to the introduction to clarify that one of the novel contributions of the paper is the generalization of previous intuitions, largely based on work on mazes, to open field environments.

I did not find it helpful that the paper is framed as evaluating different possible neural representations of uncertainty, and I don't think the rejection of the product or DDC schemes for doing this necessarily tells us much about whether or not they are used in situations where the brain might in fact represent a probability distribution.

We did not aim to suggest that our results exclude the viability of the product or DDC schemes in other brain regions or at different temporal scales. We clarified this point in the discussion (line 556).

Reviewer #3 (Recommendations for the authors):Please provide more details how the increased diversity prediction of the DDC was tested. How is the diversity decoded from the population? What do the cumulative probabilities on Figure 5 subplot b show exactly and why and how can they show the increased diversity? The authors may want to provide some intuition/more explanation about what the decoded SD reflects in Figure 5 b, d and why it reflects the diversity and not the uncertainty which would increase in the late cycles for the rest of the models too.

We rewrote the text (lines ~264-273) and the legend of the corresponding figures to clarify that we use the decoded SD and not the diversity across neurons to test the DDC code, and that the CDFs are calculated across theta cycles. We summarise the answer to the question below.

In the DDC scheme the firing rate of neurons is defined by the overlap between the target distribution and their tuning functions. Intuitively, this scheme leads to a code where the diversity of the coactive neurons increases with uncertainty. This increase in the diversity provides a useful intuition about how neuronal activities change with uncertainty in this scheme. We did not aim to decode diversity nor to directly evaluate how diversity changes within a theta cycle. Instead we aimed at directly decoding the summary statistics of the probability distribution that had been encoded in the population activity. To this end, we used the standard deviation (SD) of the distribution to quantify the uncertainty (large SD = large uncertainty, implying large diversity as well).

To achieve this, we built a maximum likelihood decoder that finds the most likely parameters (mean and SD) of the encoded distribution. We applied this decoder to early, mid, and late spikes in each theta cycle recorded either using simulated data (Figure 5B-d) or experimental data (Figure 5e-g). The plots in Figure 5b and e show the results of this decoding process. Specifically, the three curves in Figure 5b (some of them are overlapping) show the distribution of the standard deviation values decoded from the observed spikes in the three different theta phases (early, mid and late) in the form of cumulative distribution functions (CDF). An increased uncertainty of the encoded distribution would be reflected by a rightward shift of the distribution. Importantly, the three CDF curves completely overlap for the sampling and the mean schemes as these schemes do not represent uncertainty in the instantaneous population activity. Instead, these schemes represent a single trajectory throughout the whole theta cycle.

Shift of CDF is only present in the DDC scheme (green), thus indicating that this measure displays specificity to DDC.

We assume all analyses ("signatures") were applied to all models but not all of the results are shown. We appreciate that the step-by-step approach eliminating one model at a time makes for a simpler story but a figure with all results would seem to be very informative: e.g. are the other signatures that contradict the product scheme?

We agree with the Reviewer that the signatures not shown could also be informative. We thus prepared a supplementary figure to show all signatures for all encoding schemes (Figure 5—figure supplement 3).

Further to the patterns observed before, our analysis also revealed that the DDC inspired decoding of SD grows within a theta cycle not only for DDC but for the product scheme as well. This can be taken as further evidence against this encoding scheme (Figure S9a-b). The EV-index is positive for the product scheme and negative for the DDC scheme (Figure S9c). We explain the positivity in the case of the product scheme by the fact that the firing rate is the lowest for the most uncertain future predictions inflating the variability at late theta cycles. In the case of DDC scheme the EVindex is negative as the distributions encoded in subsequent theta cycles are similar.

However, we would like to add a cautionary note. The EV-index and the DDC decoded SD were not designed to work consistently for all encoding schemes and thus there is no theoretically solid justification for their specificity and robustness.

For example the positivity of the EV-index in the case of the product code may not reflect true variability of the encoded locations but the large decoding noise due to the low firing rate. Indeed, the magnitude of the EV-index decreases substantially for the product code if we consider only high spike count theta cycles (Figure 5—figure supplement 3c). Consequently, the value of these measures taken on simulated datasets could be misleading and may not generalise well to the real situation. Thus we prefer to keep this figure in the supplementary material.

Presentation: we did not understand the cumulative probability panels and how they are related to the text. E.g. lines 249-252 and Figure 5b, same for Figure 6. Maybe it would be more intuitive to show the pdf instead of the cdf? But even then, the x-axis and general interpretation remains unclear to us.

To test the DDC code we assumed that a probability distribution is encoded in each moment by the activity of the neuron. We decoded the moments (mean and SD) of this distribution in each theta cycle for the three theta phases (early, mid and late) separately using a maximum likelihood decoder. As a result of this decoding process we obtained three distributions of the decoded SD values (three SD values for each theta cycle, corresponding to early mid and late phase) and plotted these distributions separately as three CDF curves with the SD on the xaxis. The shift of these curves along the x-axis indicate differences in the magnitude of the represented uncertainty.

The interpretation of the CDF curves in Fig6b,d,h are similar.

We decided to show the distribution using the CDF as relatively small, but significant differences between the distributions are more apparent in the CDF than in the PDF (Figure R1).

We rewrote the description of these panels in the text and in the legend to clarify that CDFs are calculated across theta cycles (line ~278).

Line 333: How do the results suggest efficient planning in the hippocampus? It suggests probabilistic, i.e. close to statistically optimal computations, but the authors should provide more details why they think it is also efficient.

In the revised manuscript we clarified our wording and use the word *efficient* solely when we refer to anti-correlated samples.

[Editors’ note: further revisions were suggested prior to acceptance, as described below.]

The manuscript has been improved but reviewer #2 has one remaining issue that needs to be addressed, as outlined below.

We conducted additional simulations and analysis to address the remaining comment of reviewer #2. Our results demonstrated that our analysis is robust to the way we generate synthetic data and thus the signatures we derived can be used to discriminate the alternative coding schemes. The results of this analysis in included in the new version of the paper.

We provide a detailed answer to the Reviewers’ comments below.

Reviewer #2 (Recommendations for the authors):In this revision, the authors have made a number of improvements to what was already a systematic, rigorous examination of an important issue. The control analysis ruling out excess variance is due to different place maps across navigation-to-goal and random foraging further enhances confidence in the results, and is additionally supported by a similar analysis in Brad Pfeiffer's paper that just came out (PMID: 35396328).

We thank the Reviewer for drawing our attention to this recent paper which we included in our reference list.

My one remaining hesitation with this paper is the one I brought up in my previous review but perhaps didn't explain clearly, so I will try again. My understanding of the core logic of the paper is that the authors first establish what the signatures are of the coding schemes they wish to distinguish, by implementing these schemes in a series of models that generate synthetic spiking data. Then, they test to what extent those signatures exist in real data, and draw inferences from comparison with the simulated results.I think this is a powerful approach and am completely on board with it. However, it does raise an overall question of how robust the simulation results are: what components of the simulation are necessary and sufficient for the results? How sensitive are the results to specific parameter choices? I appreciate and agree with the authors' argument that the Kalman filter is an appropriate and relatively minimal way to model the animal's uncertainty about its own location. But it is not obvious to me what modeling the animal's uncertainty about its position contributes to the results in the first place. Would you get the same simulated signatures if all you had was a probability distribution of expected future trajectories given true current location, and then applied the various coding schemes (encoding MAP trajectory, sampling, etc)?

The reviewer suggested that representation of uncertainty associated with the current position of the animal might not be necessary to interpret experimental data. Indeed, the key to our analysis is the change in uncertainty along the planned trajectory of the animal that is less sensitive to the level of uncertainty at any given time. Accordingly, the trajectory distribution that is using the model-free alternative, which lacks the representation of current uncertainty is expected to produce similar results than the generative model that explicitly represents uncertainty in current beliefs. To corroborate our expectations, we designed a minimalist approach to generate hypothetical future trajectories without explicitly modelling the animal’s subjective uncertainty and demonstrated that our signatures robustly discriminate the different coding schemes.

Hence, my suggestion of using the SR to obtain such trajectory distributions from the animal's behavioral data, but really any approach that estimates these distributions from the data would help address my concern. This feels important, because comparison between these different generative models would give a sense of what data sets/behavioral tasks/task conditions should show the predicted signatures. Ultimately we don't want to just understand what happens in the Pfeiffer and Foster 2013 data set the authors analyze, but have some idea of what to expect say, in the dark with high uncertainty about current location, or on armed mazes where possible futures are highly constrained.

We thank the reviewer for clarifying this idea. Indeed, we agree that paper benefits from the demonstration that the signatures of coding schemes are not sensitive to the specific properties of the generative model of motion trajectories.

Therefore we designed a minimal, model-free approach to obtain trajectory distributions directly from the animal’s behavioral data. Specifically, we collected trajectory distribution for a particular target point on the real motion trajectory we first sampled multiple real trajectory segments with similar starting speed. Next, we aligned the start of these trajectory segments with the start of the target trajectory by shifting and rotating the sampled segments. This alignment resulted in a bouquet of trajectories for each actual point along the motion trajectory that outlined an empirical distribution of hypothetical future positions and speeds (Figure 7—figure supplement 2a).

Next we used the empirical, model-free distributions of motion trajectories to generate spiking data using the four different coding schemes in the same way as we did it for model-based trajectories. Next, we repeated identical analyses that were performed with the original, model-based synthetic trajectories. We found that the population gain, the increase in the DDC-decoded SD and the sampling index were potent signatures to discriminate these alternative encoding schemes (Figure 7—figure supplement 2c-g). We concluded that our results are robust against changes in the way distributions of planned trajectories are constructed. We present the results of this new analysis in a novel subsection of the Results (Analysis of signatures using model-free trajectories, line 430) and in a novel Supplemental Figure (Figure 7—figure supplement).

I realize that the paper is already extensive and thorough, but unless I'm off base with this intuition or misunderstand the authors' argument, it could actually simplify their paper if the same results obtain with a simpler way of generating probability distributions over trajectories. If they don't obtain, then this is important to point out, because of the resulting prediction that theta sequences ought to have different spiking statistics in high vs low uncertainty-about-current-position conditions.

We had a hard time deciding how to include the new, model-free analysis into the paper. We decided to present it at the end of the Results section as a separate subsection and on a new Figure Supplement. Our choice was based on the following arguments. First, we consider the presentation of the generative model of planning an insightful ingredient of the paper, as the readers are provided with new ways of thinking about hippocampal theta sequences and therefore has the perspective to inspire the design of new experiments in the future.

1. Experimental data suggests that the hippocampus is involved in model-based planning. Our original approach, implementing a model based method to generate synthetic trajectories, is thus more similar to what might be actually implemented by the hippocampal circuitry than the new, model-free alternative.

2. We believe that representing subjective uncertainty in a self-consistent way using a probabilistic model is an important principle used in many artificial systems. This idea has been also applied recently to neuronal systems and gained support from a wide range of studies, albeit from different domains of cognition than the one studied here.

3. Our generative model was a key component of our framework providing crucial insights about the variability of hippocampal theta sequences. In particular, we realized that the subjective uncertainty of the animal is an important and often neglected factor shaping hippocampal representations. In our work, this is best reflected in the relationship between the decoding error, the subjective uncertainty, and the expected change in the posterior mean that allowed us to derive the sampling index and in the way we can explain the source of the bias in the cycling index (Figure 7e).

Second, we decided to present the keep the original logic of the paper and present the main findings first using the original, generative model and only describe the model-free approach at the end of the manuscript as a control simulation. Alternatively, we could have introduce the two approaches in parallel around Figure 3 and demonstrate all results using both alternatives, but we were afraid that this would break the narrative of the main text and confuse the readers.

Reviewer #3 (Recommendations for the authors):The authors have addressed all my concerns. They have added an interesting new analysis as the result of the reviewer suggestions. Congratulations to the authors for an impressive piece of work.In the final version, I'd like to encourage the authors to better explain the cause of the bias towards similar directions described in line 404. I didn't understand it. Please elaborate.

We thank the Reviewer for their encouraging words.

We rewrote the corresponding section of the Results (lines 402-409) and the caption of Figure 7e to clarify the source of this bias.

The bias can be best understood if we consider that the true position of the animal in room coordinates that we can directly observe can sometimes be different from the animal’s own position estimate. Although this difference may be quite small (it is typically not larger than a few centimeters in our synthetic dataset in a 2×2 m arena), the important point is that this subjective position represent the animal’s best guess about its own position and hypothetical trajectories will alternate around this subjective position during generative cycling. Even if the subsequent trajectories are alternating around the subjective position they might fall into the same side of the true position if the two are different (Figure 7e).